# Pervasive epistasis exposes intramolecular networks in adaptive enzyme evolution

Karol Buda [1], Charlotte M. Miton [1] & Nobuhiko Tokuriki [1] ✉

Enzyme evolution is characterized by constant alterations of the intramolecular residue networks supporting their functions. The rewiring of these network interactions can give rise to epistasis. As mutations accumulate, the epistasis observed across diverse genotypes may appear idiosyncratic, that is, exhibit unique effects in different genetic backgrounds. Here, we unveil a quantitative picture of the prevalence and patterns of epistasis in enzyme evolution by analyzing 41 fitness landscapes generated from seven enzymes. We show that >94% of all mutational and epistatic effects appear highly idiosyncratic, which greatly distorted the functional prediction of the evolved enzymes. By examining seemingly idiosyncratic changes in epistasis along adaptive trajectories, we expose several instances of higher-order, intramolecular rewiring. Using complementary structural data, we outline putative molecular mechanisms explaining higher-order epistasis along two enzyme trajectories. Our work emphasizes the prevalence of epistasis and provides an approach to exploring this phenomenon through a molecular lens.

Enzyme evolution proceeds via the stepwise accumulation of adaptive, neo-functionalizing mutations. Since enzyme functions are supported by sophisticated three-dimensional structures, underpinned by highly connected amino acid networks, functional optimization often involves the rewiring of these intramolecular networks by adaptive mutations (Fig. 1a)[1]. Studying these mutations through the lens of intramolecular networks has yielded a deep mechanistic and structural understanding of enzyme evolution[1–7]. For example, by probing non-additive interactions between two mutations, *i.e.*, epistasis, one can unveil key residue interactions that enhance or compromise the optimization of an enzyme's function (Fig. 1b,d). However, in strongly intertwined intramolecular networks, multiple residues concurrently interact, and the introduction of further mutations is likely to rewire previously established interactions, giving rise to higher-order epistasis (interactions between three or more mutations)[8] (Fig. 1e). Consequently, pervasive higher-order epistasis can cause mutational and epistatic effects to appear highly idiosyncratic, that is, exhibiting variable effects depending on the background in which they occur[9] (Fig. 1b-e). Idiosyncrasy challenges our understanding of the mechanistic and functional basis of mutations because, when epistasis is widespread, the characterization of single mutational- and epistatic-effects in a particular genetic background may not accurately reflect the contribution of these effects during adaptive evolution. Thus, a robust description of the extent of idiosyncrasy, stemming from higher-order epistasis, is required if we wish to understand the topology of intramolecular networks and enzyme evolution.

While epistasis has been extensively described over the last decade, the degree to which it creates apparent idiosyncrasy in mutational- and epistatic- effects during adaptive enzyme evolution remains unknown. Several studies have systematically characterized epistasis in organisms[8,10] and found extensive idiosyncrasy among genomic mutations[9,11]. These interactions between genomic mutations are often rooted in complex phenomena, *e.g.*, arising from a non-linear relationship between the organismal fitness and the function of a protein, genetic dominance[12], or metabolic network dynamics[13]. Yet, global or non-specific epistasis, the non-linear relationship between a protein biochemical trait and a measurable phenotype, could also account for the majority of idiosyncrasy, even in single protein studies[14–17]. Indeed, models that simply correct for non-linear mapping account for >90% of the phenotypic variation across proteins[16–18]. Notably, these studies were all conducted on random, non-adaptive mutations – in contrast, recent work on adaptive trajectories at the single enzyme level has shown that specific epistasis, *i.e.*, epistasis originating from biophysical interactions between amino acids,

[1]Michael Smith Laboratories, University of British Columbia, Vancouver, Canada. ✉e-mail: tokuriki@msl.ubc.ca

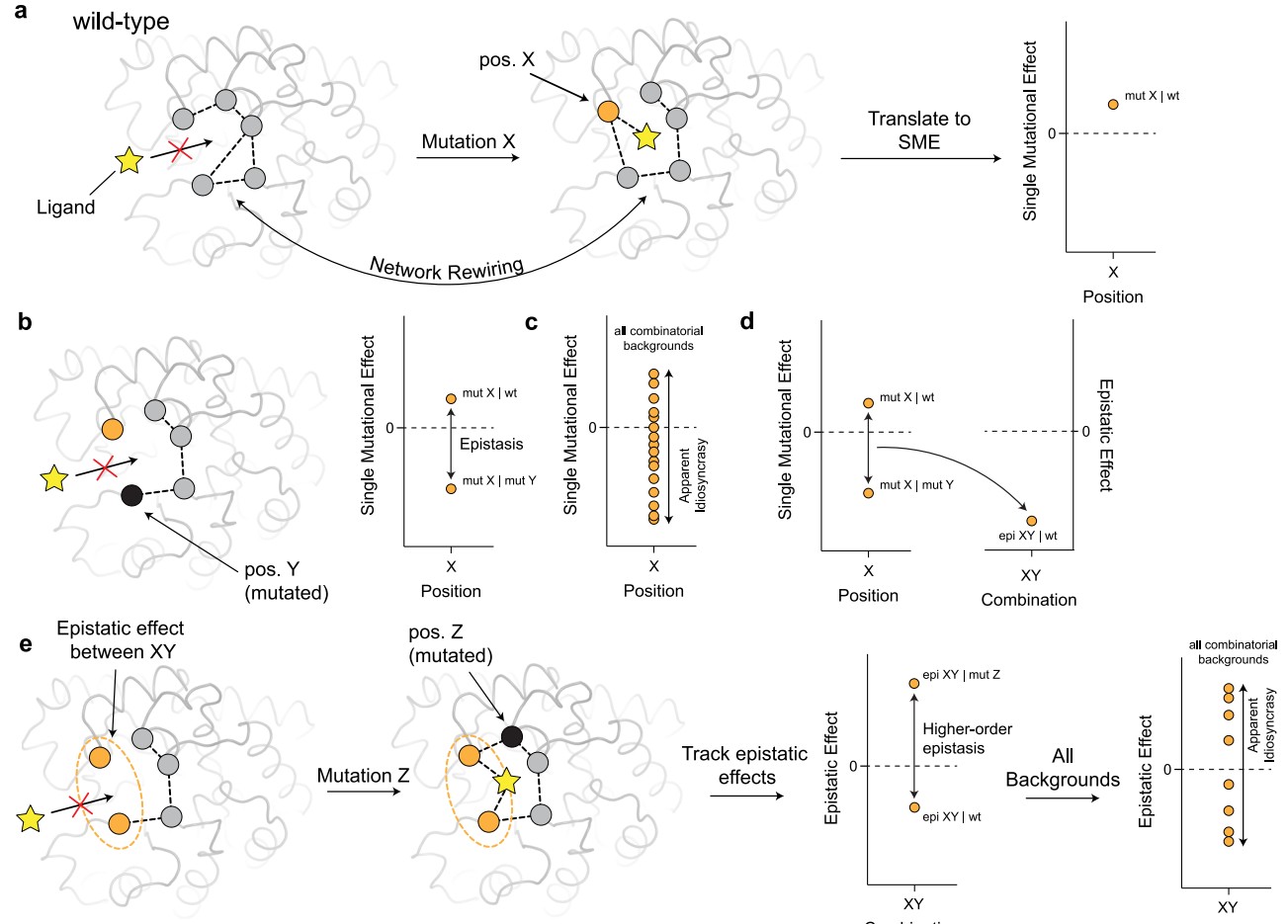

**Fig. 1 | Intramolecular network rewiring underpins apparent idiosyncrasy in mutational effects. a** A ligand (yellow star) cannot bind in the active site of a wild-type (wt) putative enzyme until a mutation (orange node) at position X (pos. X) rewires the entire network, unlocking activity toward the ligand. The change in function upon mutation at pos. X can be plotted as the single mutation effect (SME) in the wt background. **b** Mutation at pos. X does not enhance ligand stabilization if pos. Y (black node) has been previously mutated, due to an alternative intramolecular network pre-existing in this background. When the SME is plotted in this background, the discrepancy between SMEs represents epistasis. **c** When pos. X is mutated in a variety of backgrounds, epistasis will cause this mutation to appear highly idiosyncratic. **d** Epistasis between pos. X and Y can be plotted as an epistatic effect in the wt background. **e** The epistatic effect between pos. X and Y (orange node and halo) differs when pos. Z (black node) is mutated, a product of higher-order epistasis, and appears idiosyncratic across various other backgrounds.

prevails[2,19,20]. Therefore, it stands to reason that the patterns and prevalence of idiosyncratic epistasis in adaptive enzyme evolution may substantially differ from large-scale, random mutational sampling studies. Furthermore, idiosyncratic effects may also arise from a more tractable phenomenon: the unique alteration of molecular interactions embedded in intramolecular networks.

To date, conventional approaches aimed at deconvoluting epistasis by either (*i*) capturing non-specific epistasis across the landscape[21], (*ii*) quantifying mean epistasis using background-averaging methods[22], or (*iii*) by isolating a few specific mutants for biophysical characterization of the mutational interactions[23]. The first and second approaches typically require a fully annotated, experimentally measured, combinatorial landscape, *i.e.*, a fitness landscape encompassing all mutational combinations of a small set of mutations. Combinatorial landscape data are then (*i*) transformed onto a linear scale using non-linear transformation techniques[16–18,21] and/or (*ii*) analyzed using variations of the Walsh-Hadamard transform[8,22]. They are then distilled down to epistatic coefficients that represent the average mutational- and epistatic- effects of all mutations across every genotypic background present in the landscape. While these approaches excel at recapitulating landscape-wide epistatic trends, they can mask the contribution of specific interactions in select, evolutionarily

relevant genotypes, therefore complicating the identification of key amino acid connections within the intramolecular network of an evolutionary intermediate. The third approach examines, in greater detail, the specific biophysical interactions existing within intramolecular networks, by measuring epistasis in a reference genotype (usually the wild-type background) and interpreting these effects with structural evidence[1,6,24–26]. However, these studies generally focus on a single, or only a few, background(s), generally devoid of higher-order epistasis quantification, and rarely examine whether these interactions are rewired – or distinctly wired – in different genotypes. Thus, to comprehensively explore intramolecular networks in enzymes and their rewiring during adaptive evolution, it is pertinent to develop a hybrid approach, wherein apparent idiosyncrasies can be captured across a combinatorial landscape and changes in epistasis can be tracked to the molecular level to expose novel, rewired, interactions.

What are the patterns and prevalence of epistasis in adaptive enzyme evolution? How can higher-order epistasis and seemingly idiosyncratic effects be used to explore the rewiring of intramolecular networks underlying enzyme adaptation? In this study, we address these questions by comprehensively quantifying and extracting epistasis and higher-order epistasis from 41 combinatorial landscapes, spanning seven different enzymes. We found that, in adaptive

**Table 1 | Combinatorial landscapes analyzed in this study**

| Enzyme | No. of mutations | Conditions | No. of total measurements | Measured trait | Reference |
|---|---|---|---|---|---|
| OXA-48[a] | 4, 6, 6 | 2, 2, 2 | 288 | $IC_{50}$ | 26 |
| TEM-1[b] | 5, 4 | 1, 11 | 208 | MIC, Growth rate | 31,32 |
| AP | 5 | 1 | 32 | $k_{cat}/K_M$ | 22 |
| NfsA[c] | 7, 7 | 1, 1 | 256 | $EC_{50}$ | 27 |
| DHFR[d] | 4, 6, 5, 5 | 5, 1, 2, 2 | 272 | $IC_{50}$, $IC_{75}$, $k_{cat}/K_M$, $K_i$ | 28–30 |
| MPH[e] | 5 | 8 | 256 | Lysate activity | 19 |
| PTE[f] | 6 | 2 | 128 | Lysate activity | 25 |

[a]Three independent trajectories each probed using two inhibitors
[b]Ref. 32 explored 15 inhibitors for the same set of four mutations; 4 landscapes were removed leaving 11 conditions
[c]Two separate mutational trajectories for the same enzyme and substrate
[d]Ref. 29 explored four mutations using five different substrates; ref. 30 explored both $k_{cat}/K_M$ and $K_i$ for two mutational trajectories
[e]Ref. 19 explored the same five mutations under eight different metal environments
[f]Six mutations were explored using two substrates, one in ref. 25 and one outlined in this study (see Methods)

trajectories, over 94% of mutational and epistatic effects appear highly idiosyncratic, rendering them highly context-dependent. Furthermore, we showed that such underlying idiosyncrasy heavily impaired functional predictions of evolutionary intermediates and endpoints. By exploring evolutionarily connected genotypes, we demonstrate that changes in epistasis expose network rewiring events. Finally, we feature two examples of higher-order intramolecular network rewiring and analyze epistatic effects in the light of existing structural data, proposing putative molecular mechanisms that may drive network rearrangements. Our work serves as a barometer for estimating the extent of apparent idiosyncrasy in single mutational- and epistatic-effects in enzyme evolution and constitutes an approach for exploring the rewiring of biophysical interactions within the intramolecular networks that support enzyme functions.

## Results

### Statistical characterization of 41 combinatorial landscapes

We first conducted a literature search of combinatorially complete fitness landscapes of enzymes. We limited our search to studies probing only single mutations per position. They were then filtered to ensure that the landscapes explored four or more positions ($n \geq 4$) and functionally characterized all possible combinations of these mutations ($2^n$ variants). From these, we only retained studies reporting enzyme function as a continuous variable. Using these cut-offs, we obtained a working set of ten studies, exploring seven different enzymes (Table 1). Some of these mutations were accumulated during directed evolution or enzyme engineering toward a novel function (phosphotriesterase, PTE; β-lactamase, OXA-48; nitroreductase, NfsA)[27–29]. Others were identified from naturally occurring evolutionary trajectories, either through a retrospectively identified path using ancestral sequence reconstruction (methyl-parathion hydrolase, MPH)[19,20], the presence of clinically relevant mutations (dihydrofolate reductase, DHFR, and β-lactamase, TEM-1)[30–34], or in the case of alkaline phosphatase (AP), by using previously characterized active site mutations[24]. The final dataset consisted of 56 unique mutations; we ensured that the majority (54) was located within the protein open reading frame, but retained two mutations in the promoter region (in DHFR and TEM-1)[30,33]. We also removed three TEM-1 landscapes probing growth rates for AMP, AMC, CAZ, and TZP as these data yielded binary fold-change values after performing the non-linear transformation (see Methods for details). These data were analyzed as 41 separate combinatorial landscapes, totaling 1,440 genotype-phenotype data points. Note that some studies explored the same set of positions in a unique enzyme, albeit with a different substrate, inhibitor, or metal cofactor (Table 1 and Supplementary Data 1).

Next, the mutational data were processed to allow for a streamlined analysis using our computational pipeline (see Code Availability). Trajectories that explored different mutational combinations for the same

enzyme were treated as separate combinatorial landscapes, as were the combinatorial landscapes characterizing the function of the same subset of mutants across different substrates, ligands, or metals. Due to the variety of measured functions, ranging from direct physicochemical properties of enzymes to indirect effects on the cellular phenotype, all enzyme functions were normalized relative to their wild-type (wt) background, providing a fold-change in enzyme function ($F$), which was then log-transformed (see Methods). Replicate measurements were transformed to a single mean value, before normalization and $\log_{10}$ transformation. Not accounting for the wt genotypes where $\log_{10}(F) = 0$, we obtained 1,399 $F$ values for further analysis (Supplementary Data 2). We then applied a four-parameter non-linear transformation to our dataset to remove the influence of non-specific epistasis (Methods and Supplementary Data 3). Finally, we also carried out all analyses on a reduced dataset to monitor statistical artefacts originating from potential cross-landscape correlation (see Methods).

### Heterogeneity in single mutational effects

To paint a comprehensive picture of the prevalence of epistasis in the selected enzymes, we first extracted the functional effect of every single mutation at a given position, across all available genetic backgrounds. The data were processed to provide the single mutational effect (SME) of a given mutation across every genotype (see Methods). In a combinatorially complete fitness landscape of $n$ mutations, a particular mutation occurs in $2^{n-1}$ distinct genetic backgrounds, hence, each combinatorial landscape contains $n \times 2^{n-1}$ SMEs. Using this approach, we collected the 3,936 SMEs (Fig. 2a) and faceted them by the 198 unique mutation- substrate/inhibitor/metal pairs, simply referred to as positions – e.g., the effect of a mutation at the same amino acid residue for two different substrates is treated as a different position (Supplementary Data 4). We chose to use a significance threshold of 1.5-fold for all analyses; this was the median error rate (calculated as two standard deviations) for all replicate measurements available in our dataset. Using this threshold, we found that the sign of the SMEs was 18% negative, 30% neutral, and 52% positive across all genotypes. This constitutes a relatively even split, despite a slight bias toward a positive effect, consistent with the fact that most mutations were adaptive and originally identified due to their beneficial effect on function (Fig. 2a).

Next, we characterized the heterogeneity – the spread in effects across different backgrounds – in SMEs for a given position by plotting and analyzing the variance in the distribution of SMEs at each position (Supplementary Data 5). The spread in SMEs across different genetic backgrounds demonstrates the presence of significant context-dependence; accordingly, the degree of SMEs scatter for a position across various genotypes reflects the existence of epistasis in pairwise, and possibly higher-order, interactions, and provides a metric for the apparent idiosyncrasy that a position exhibits. To capture the spread, we computed two standard deviations (2 SD) for the SMEs at each

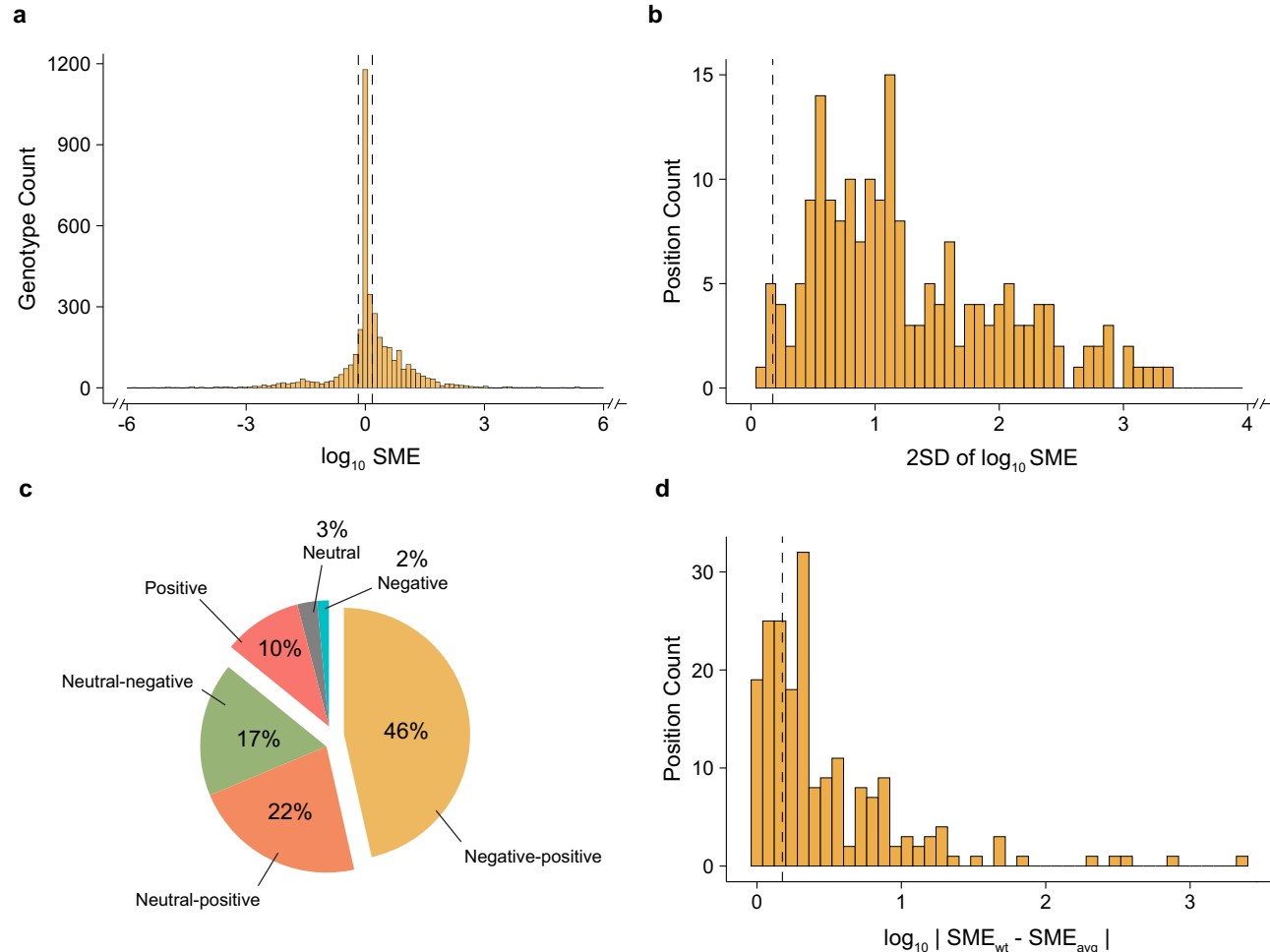

**Fig. 2 | A quantitative survey of single mutational effects. a** Distribution of single mutational effects ($\log_{10}$ SME) at every position across all genotypes for every combinatorial landscape. **b**, Distribution of the spread in the SMEs (2 standard deviations, or 2 SD, of $\log_{10}$ SME) at each of the 198 positions. **c**, Categorisation of the sign-changing behavior in SME distribution for each of the 214 mutations.

**d**, Distribution of the absolute difference between wt and avg SME for each of the 198 positions. Data omitted due to x-axis scaling in **a** and **b** can be found in Supplementary Fig. 1. Dashed lines represent the 1.5-fold significance thresholds. Source data available in the Source Data file.

position, which should encompass ~95% of the functional variability and inform on the extent to which epistasis distorts SMEs across each background relative to the mutation's average effect. We deemed positions to be highly heterogeneous when their 2 SD > 1.5-fold (our threshold metric). We found that idiosyncrasy prevails in our dataset: 96% (189/198) of positions exhibit significant spread (Fig. 2b and Supplementary Table 1). Many positions exhibit much stronger heterogeneity than the threshold: 71% (141/198) show a 2 SD > 5-fold, and 56% (110/198) show a 2 SD > 10-fold (Fig. 2b and Supplementary Table 1). The reduced dataset exhibited similar, albeit slightly lower, values (Supplementary Table 2). We also characterized how the magnitude of the spread in SMEs affects their sign-changing behavior, *i.e.*, whether the sign of the SME varies between positive, negative, and/or neutral in different genotypic backgrounds. Only 14% of the positions retained the same sign across all genotypes while 17% of positions showed neutral and negative SMEs and 22% of positions displayed both neutral and positive SMEs. Interestingly, 47% of positions showed background-dependent variability between positive and negative effects (Fig. 2c and Supplementary Table 3). The proportion of positive *versus* negative SMEs for each position within this category varied, indicating that the sign contribution was indeed heterogeneous across different backgrounds (Supplementary Fig. 2). However, we found that the reduced dataset showed a significantly lower proportion of negative-positive positions (Supplementary Table 4), contrasting that of the larger

dataset, which we elaborate on in the discussion. Finally, we sought out the proportion of SMEs that are well represented by the average SME at a given position across the entire landscape. We began with the wt background SME ($SME_{wt}$) and computed the absolute difference between the $SME_{wt}$ and the mean SME ($SME_{avg}$) for every mutation across all representative genotypes (Fig. 2d). This approach captures the apparent idiosyncrasy of each position by measuring the extent to which the SME at a particular genotype reflects the mean effect. For 67% of positions, the $SME_{wt}$ deviates from $SME_{avg}$ by >1.5-fold (Fig. 2d **and** Supplementary Table 5), while the sign of the $SME_{wt}$ remains similar to that of $SME_{avg}$ (only 1% of positions show a significant difference in sign). This proportion did not only apply to $SME_{wt}$ but also to the SME in every other genotype – 58% (2283/3936) of positions showed a deviation >1.5-fold between the SMEs across every genotypic background and their $SME_{avg}$ (Supplementary Table 7). These statistics remained largely similar in the reduced datasets (Supplementary Tables 6 and 8). Together, these observations suggest that, in any given adaptive landscape, mutations appear highly idiosyncratic: there is a 90–96% chance that a given mutation will exhibit epistasis with other adaptive mutations, and in 23–47% of cases the effect of a mutation will vary between beneficial and detrimental, depending on the presence of other adaptive mutations, while in 66–70% of cases the SME observed in the wt background (or 55–58% for any given background) will be poorly representative of its effect in other backgrounds.

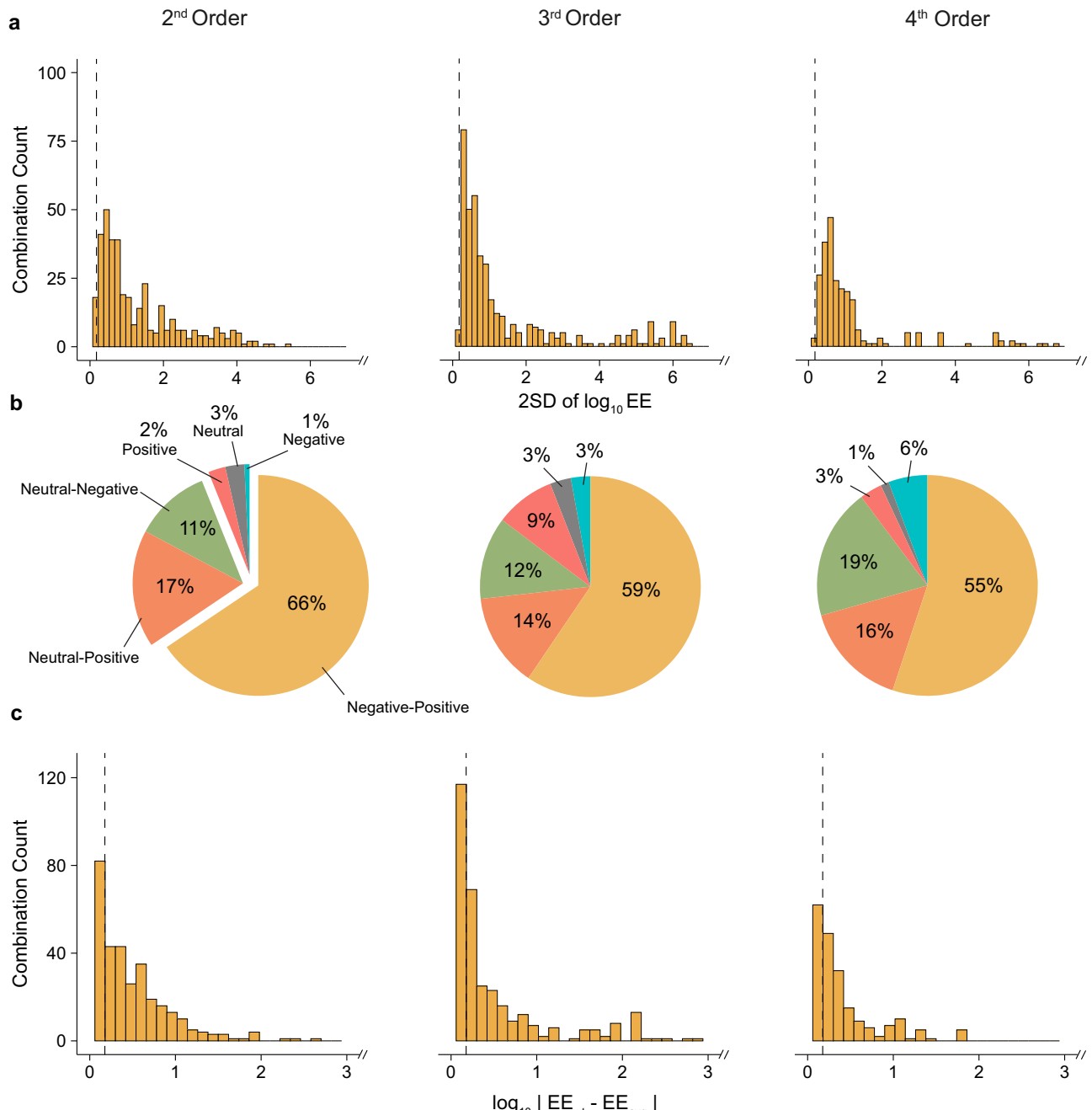

**Fig. 3 | A quantitative survey of epistatic effects. a** Histogram of 2 standard deviations (2 SD) of $\log_{10}$ epistatic effects (EEs) at each order for every mutational combination. Data omitted due to x-axis scaling can be found in Supplementary Fig. 3. **b** Categorisation of the EEs at each order for all mutational combinations by sign. **c** Distribution of the absolute difference between wt EE and avg EE at each combination. Data omitted due to x-axis scaling can be found in Supplementary Fig. 3. Dashed lines represent the 1.5-fold significance thresholds. Source data are available in the Source Data file.

## Heterogeneity in epistatic effects

The extensive epistasis that creates heterogeneity in SMEs stems, at the very least, from strong and pervasive pairwise epistasis – but does it end here? We expanded our survey of idiosyncrasy from SMEs to mutational interactions. To this end, we calculated the epistatic effects (EEs) for every genotype at any given order by computing the deviation between the observed function and the predicted function using the sum of all constituent SMEs and lower-order EEs (see Methods). By taking these data and utilizing the same metrics of spread as with the SME analysis, we can detect the presence of higher-order epistasis and its ability to create apparent idiosyncrasy across epistatic effects.

Compared to the 198 surveyed positions, we extracted 395 pair-wise-, 422 three-way-, and 293 four-way- combinations. Since 18 of the 41 landscapes only probed 4 mutations, 18/263 four-way combinations could not be examined as they only occurred in one genetic background, leaving 245 four-way combinations viable for analysis. Overall, we collected 8,658 values for EEs from the 2nd to the 4th order (Supplementary Data 6). Surprisingly, we found that nearly all pairwise combinations (95% or 376/395) show significant idiosyncrasy in EEs (Fig. 3a). Although this proportion decreased at less stringent thresholds, even with a 10-fold significance cut-off, 44% of pairwise combinations retain a significant spread (Supplementary Table 1). Interestingly, this trend remains consistent at higher orders: 94% (397/

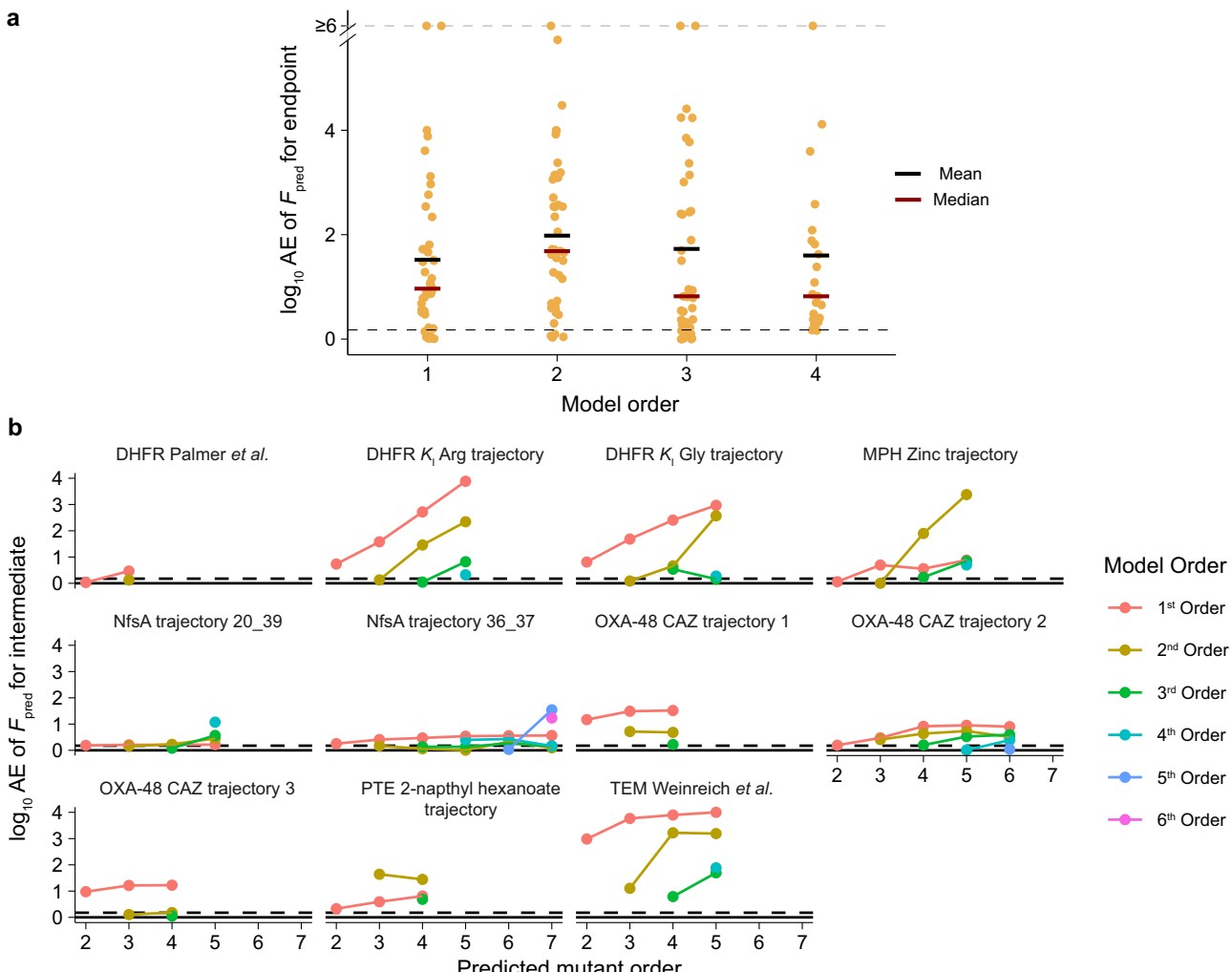

**Fig. 4 | Apparent idiosyncrasy distorts functional predictions. a** $\log_{10}$ of the absolute error (AE) of the predicted function ($F_{pred}$) at the 1st, 2nd, and 3rd orders ($n = 45$), and the 4th one ($n = 23$) using the biochemical model. The means and medians of the absolute error are shown as black and red bars, respectively. The 1.5-fold significance threshold is depicted as the lower dashed line. **b** $\log_{10}$ of the absolute error (AE) of the predicted function ($F_{pred}$) for each genotype along the most accessible path in 11 adaptive landscapes, indicated by subtitles. Predicted mutant order indicates the mutational order of the predicted genotype (relative to the WT). Colors indicate which order of the biochemical model was used. Source data are available in the Source Data file.

422) of three-way combinations, and 99% (244/245) of the four-way combinations exhibit significant heterogeneity (Fig. 3a). The impact of spread in EEs was also apparent in their sign variability – 66% (259/395) of combinations show pairwise EEs that can be either positive or negative, depending on the background (Fig. 3b). Proportions of positive-negative combinations appear similar to that of the SMEs (Supplementary Fig. 4). Only a handful of combinations (6%) exhibit single sign EEs, and the remaining combinations were either neutral-negative (11%) or neutral-positive (17%). For higher-order combinations, positive-negative sign variability remains high: 60% for three-way, and 55% for four-way (Fig. 3b and Supplementary Table 3). Like SMEs, the wt background EE ($EE_{wt}$) showed a high deviation from the mean ($EE_{avg}$) for each combination. More than half of pairwise (60% or 235/395), 52% (or 218/422) of triplet, and 60% (or 148/245) of quadruplet $EE_{wt}$ deviated from $EE_{avg}$ by more than 1.5-fold (Fig. 3c and Supplementary Table 5). This high deviation was also representative of other non-wt backgrounds (Supplementary Table 7). Interestingly, more combinations showed significant sign discrepancy in $EE_{wt}$ *versus* $EE_{avg}$ than in SMEs. Although still in the minority, 9% (37/395) of pairwise-, 6% (26/422) of three-way-, and 7% (17/245) of four-way- interactions had a significantly different sign effect in the $EE_{wt}$ than the $EE_{avg}$. Again, all results remained comparable upon removal of potentially

correlated landscapes (Supplementary Tables 2, 4, 6, 8), including the trends in sign heterogeneity, unlike the SMEs. These high levels of apparent idiosyncrasy in EEs, similar to the ones observed for SMEs, imply that higher-order epistasis is equally prevalent and influential compared to lower-order epistasis and that epistatic effects continue to appear idiosyncratic even at higher orders.

### Apparent idiosyncrasy confounds prediction

The strong prevalence of heterogeneity in SMEs and EEs raises the question of the extent to which seemingly idiosyncratic effects can be used for functional prediction across the landscape. We computed the predicted functions ($F_{pred}$), by using the biochemical model (utilizing $SME_{wt}$ and $EE_{wt}$)[22]. The biochemical model examines predictability given a limited amount of mutational data, *i.e.*, mutational effects in the wt background, thereby directly utilizing each context-dependent effect as a model coefficient. We gradually increased the accessibility to higher mutational orders used for prediction, up to the 4th order (Fig. 4a). To assess the functional prediction accuracy, we measured the absolute error (AE) of the predicted *versus* observed function for the endpoint, or most derived, genotype in each landscape (Fig. 4a). We found that the biochemical model showed high AEs with a marginal improvement in median AE, but not mean AE, even upon introducing 4th order epistatic

**Fig. 5 | Change in $EE_{wt}$ reveals functional patterns of network rewiring.** For every genotype with $n$ mutations (circular nodes) a node with $n − 1$ constituent mutations was identified. The $(n − 1)^{th}$ interaction is used as a reference point (blue line between square nodes), and a mutation that affects this interaction at the $n^{th}$- order (red flat-headed arrow) marks a change in epistatic effect ($EE_{wt}$; triangles). Percent frequencies of each transition category are shown, with a further break-down of a change in epistasis to 'constructive' and 'disruptive'. Blue, red, and grey represent positive, negative, and no epistasis, respectively.

information (Fig. 4a and Supplementary Table 9). This poor performance of the biochemical model demonstrates how apparent idiosyncrasy can distort functional predictions in enzyme fitness landscapes. Prediction errors were also similar in the reduced dataset (Supplementary Table 10). We also ensured that the epistasis could be effectively captured using an alternative model, namely the background-averaged model via linear regression (utilizing $SME_{avg}$ and $EE_{avg}$; see Methods)[8,10], and indeed found that prediction accuracy of the background-averaged model increased with each incorporated order, with the $3^{rd}$ order being sufficient for predicting functions within the significance threshold (Supplementary Fig. 5 and Supplementary Table 9).

To further illustrate the extent to which apparent idiosyncrasy can confound predictions, we filtered our dataset down to 11 adaptive landscapes, i.e., landscapes where the substrate or ligand is assumed to be the primary selection pressure that led to the accumulation of the probed mutations. For each landscape, we retained a single most accessible path, in which the most functionally advantageous mutation is fixed at each step. We then computed a predicted function for each intermediate genotype along the most accessible path using ascending orders of the biochemical model (Fig. 4b and Supplementary Table 11). Similar to the endpoint prediction, we found that the biochemical model failed to accurately predict the function of most genotypes at every order. Interestingly, some highly mutated genotypes along the trajectory were, in fact, better predicted by lower-order information, a phenomenon on which we will elaborate in the discussion (Fig. 4b and Supplementary Table 11). This reinforces the propensity for apparent idiosyncrasy to confound functional predictions, as well as its ability to distort inferences made on evolutionary trajectories from limited mutational data.

**Apparent idiosyncrasies reveal intramolecular network rewiring**
Given the strong prevalence of apparent idiosyncrasy in these combinatorial landscapes, we sought to explore how a context-dependent effect may reflect unique interactions within intramolecular networks. We decided to only retain combinatorial landscapes based on $k_{cat}/K_M$, $K_i$, and lysate activity, that is, functional readouts that are more likely to reflect biophysical interactions between amino acids that directly

relate to the enzymes' mechanisms. Thus, we focused our analysis on the following models: AP, DHFR, MPH, and PTE, representing 544 genotype-phenotype measurements.

We then sought to identify potential patterns of epistasis across mutational steps in these enzymes. By focusing on how EEs change upon the introduction of a new mutation, we attempted to uncover key connections within the intramolecular network between the adaptive mutations. Using the wt background as a reference, we collected the $EE_{wt}$ for every $n^{th}$-order genotype (where $n > 2$) and extracted every possible transition from an $(n − 1)^{th}$ genotype to the $n^{th}$ genotype – a total of 1027 transitions (Fig. 5). Mathematically, the $EE_{wt}$ for an $n^{th}$-order interaction describes the change in effect that a new mutation confers to any constituent $(n − 1)^{th}$-order interaction (Eq. 5 and Eq. 2 in Methods). For example, a negative $EE_{wt}$ for a three-way interaction in a $3^{rd}$ order mutant versus a pairwise interaction with a positive $EE_{wt}$ in the $2^{nd}$ order mutant suggests that the positive pairwise EE is disrupted by the presence of the new mutation (Fig. 5). In our data, we found that 23.6% (242/1027) of transitions showed no significant $(n − 1)^{th}$ and $n^{th}$-order epistasis, 14.3% (147/1027) showed no change in pre-existing epistasis, and 24.4% (251/1027) of transitions show higher-order epistasis, where none existed previously (Fig. 5). Interestingly, the remaining 37.7% (387/1027) of transitions show a change in epistasis, where the epistatic effect of a pre-existing interaction changed due to new, higher-order epistasis. These changes were further categorized: $n^{th}$-order interactions of the same sign were deemed constructive, while those of the opposite sign were marked disruptive. The minority of changes (24.0%; 93/387) were enhancing while 81.7% (371/454) were diminishing (Fig. 5). We observed similar trends for the reduced dataset (Supplementary Table 12). Expanded upon in the discussion, these data provide three key insights: (i) new mutations are often uncoupled from other pre-existing mutations or interactions that arose during adaptive evolution (in 37.9% of cases), (ii) mutations can give rise to new epistasis stemming from higher-order order connections (Fig. 5), (iii) the prevalence of changes in epistasis across evolutionarily connected genotypes may account for the low predictive power of the biochemical model (Fig. 4).

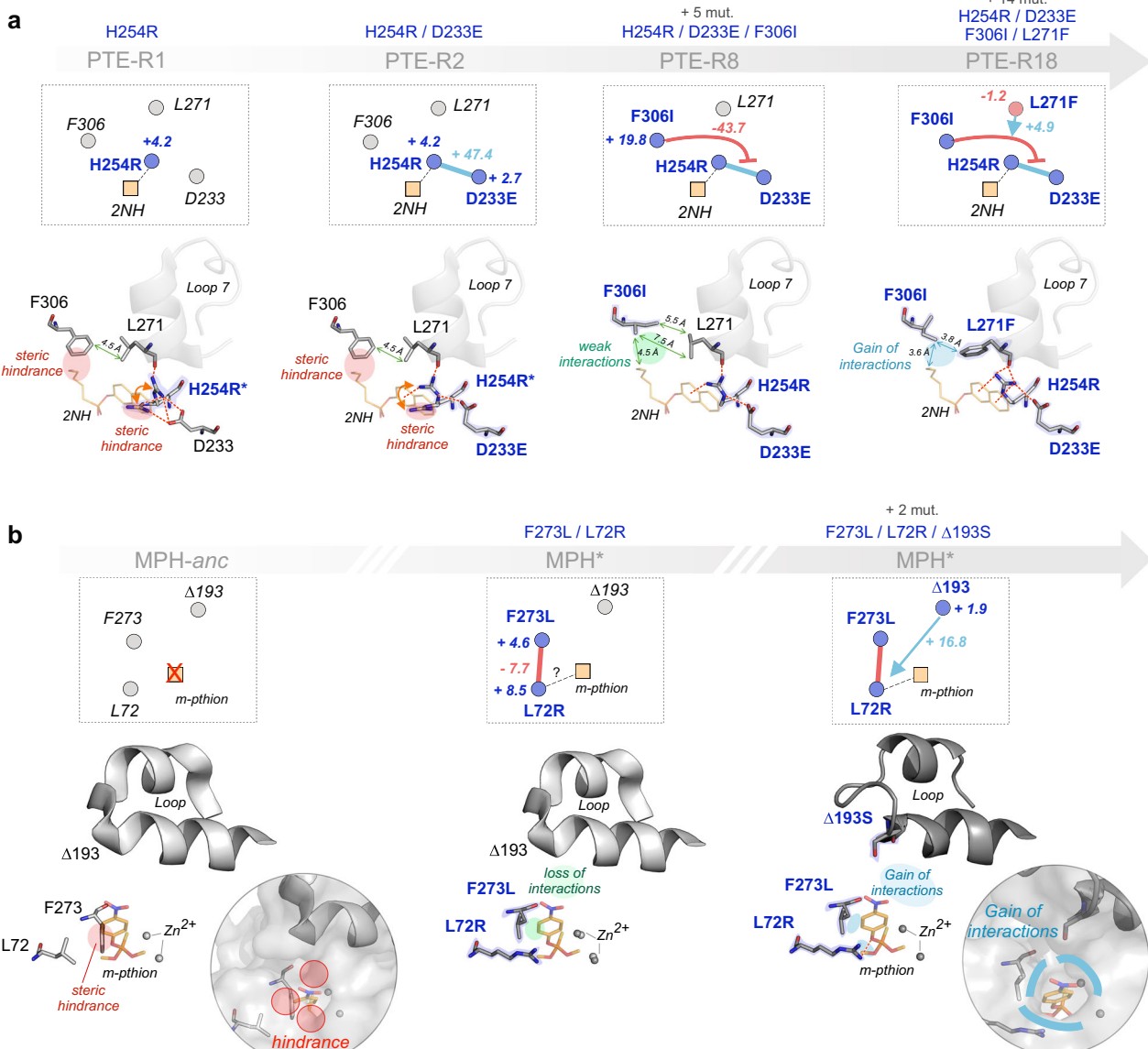

**Fig. 6 | Changes in epistasis expose intramolecular networks in enzymes. a** In PTE-R1 (PDB ID: 4xaf), H254R adopts two conformations (H254R*), one of which prevents the 2NH substrate from binding as in the most evolved variant, PTE-R18 (PDB ID:4e3t). A favorable pairwise epistatic interaction between H254R and D233E (in PTE-R2, PDB ID: 4xd5) is explained by the stabilization of the 'bent' conformer of Arg254, which enables substrate binding. Across the active site, F306I relieves a steric clash between the hexyl moiety of the substrate, but concomitantly, likely weakens hydrophobic interactions with this substituent (PTE-R8, PDB ID: 4xay). L271F strengthens the stabilization of Arg254 and Loop7, forming a robust intra-molecular network with Glu233 in PTE-R18. This cancels the negative three-way epistasis between 306-233-254 by repositioning the substrate such that Ile306 can form stronger hydrophobic interactions with the hexyl moiety, enhancing catalysis in PTE-R18. A 2NH analogue bound in PTE-R18 (PDB ID:4e3t) is overlayed in each structure to illustrate the most optimal binding orientation **b** In the MPH ancestor (MPH-anc, PDB ID: 6c2c), methyl-parathion (*m-pthion*) cannot bind in the optimal orientation obtained by molecular docking in the extant MPH* enzyme (PDB ID: 1p9e)[18], due to multiple steric clashes at the entrance of the active site, particularly with Phe273. While F273L relieves this hindrance, and L72R enhances leaving group stabilization, their pairwise interaction is negative. This may stem from the high conformational freedom of the substrate due to weaken hydrophobic interactions on the opposite side of the active site. The Δ193S insertion may compensate for this detrimental effect by tightening the active site via altered loop dynamics, thus enhancing enzyme-substrate complementarity. Note that there is no structure of the F273L/L72R mutant (middle panel), this view is an overlay between MPH-anc and MPH*.

## Molecular basis for higher-order intramolecular networks

To further demonstrate the insights gained from intramolecular network analysis, we present two examples of how changes in higher-order epistasis can be supported by structural data to provide putative molecular mechanisms for network rewiring. For instance, in PTE, the laboratory evolution of arylester hydrolysis required the initial fixation of mutations Asp233Glu and His254Arg[3]. When introduced into the wt PTE, His254Arg adopts two side-chain rotamers, one of which sterically hampers substrate binding (PTE-R1, Fig. 6a)[2,3]. The Asp233Glu mutation preferentially stabilizes the productive Arg254 rotamer, in turn positioning the substrate in a productive conformation, resulting in a 47.4−fold positive pairwise interaction (PTE-R2, Fig. 6a). The apparent idiosyncrasy of this pairwise interaction is revealed, however, upon the introduction of Phe306Ile, which drastically diminishes the synergy of the Glu233-Arg254 interaction (Fig. 6a). While Phe306Ile does not lead to strong pairwise interactions with either Asp233Glu (EE_wt -1.7−fold; Supplementary Data 6) or His254Arg (EE_wt -2.1−fold; Supplementary Data 6), it gives rise to diminishing, -43.7−fold negative three-way

epistasis. The mechanism for this apparent higher-order antagonism is likely rooted in substrate positioning. Indeed, Phe306Ile eliminates a steric clash exerted by Phe306, elongating the cavity to facilitate the binding of the hexyl moiety (PTE-R8, Fig. 6a). Since these three mutations relieve different bottlenecks, we hypothesize that their respective positive effects may vanish when combined. This may be the result of mutational incompatibility leading to increased conformational freedom of the substrate in the cavity, which decreases stabilizing interactions and potentially favors non-productive binding modes. Indeed, at higher orders, the detrimental epistatic contribution of Phe306Ile is alleviated, as the network continues its rewiring: Leu271Phe aids in repositioning Phe306Ile and Arg254, further enhancing interactions with the substrate and the active site loop 7 (PTE-R18, Fig. 6a), whose stabilization is essential for high arylesterase activity[35]. Hence, evolution overcomes this unique context-dependent, three-way incompatibility by acquiring additional stabilizing mutations, restoring the positive contribution of the Glu233-Arg254 interaction.

In the natural evolution of an organophosphatase, MPH, five critical mutations were identified as essential for organophosphate hydrolysis[19]. Note that, while the combinatorial landscapes of these mutations were evaluated in eight different metal environments[20], here we focus our analysis on the 3rd-order mutant with the highest function in the zinc environment (MPH + Leu72Arg/Phe273Leu/Δ193Ser), the putative adaptive environment. The high catalytic activity of this mutant is characterized by strong positive epistasis between Leu72Arg, Phe273Leu, and Δ193Ser. However, this three-way network stands out, due to a pairwise incompatibility between Leu72Arg and Phe273Leu, compensated for by Δ193Ser. Initially, the ancestral MPH cannot accommodate the methyl-parathion substrate in its narrow active site (Fig. 6b). The Leu72Arg mutation likely stabilizes the negative charge developing on the leaving group oxygen at the transition state, whilst the Phe273Leu seems to eliminate a steric clash with the substrate, both independently enhancing the organophosphatase activity (+8.5− and +4.6−fold, respectively). The pairwise interaction is negatively epistatic (-7.7−fold), however. The enzyme-substrate conformation observed in the MPH* crystal structure (MPH + Leu72Arg/Phe273Leu/Δ193Ser + 2 mutations) suggests a redundancy in mechanistic contribution: while Arg72 seems to promote leaving group stabilization in the productive substrate conformation, the concomitant relief of a steric clash by Phe273Leu may reposition the substrate away from Arg72, causing a loss of interactions, thus decreasing catalysis (Fig. 6b). This mechanistic bottleneck and the resulting negative pairwise epistasis are alleviated by Δ193Ser, however. This nearly neutral mutation (+1.9−fold) elicits a strong positive SME in the double mutant background (+31−fold; Supplementary Data 6), primarily stemming from the positive three-way interaction (+16.8−fold). While the precise mechanism remains unknown, the serine insertion appears to significantly increase the backbone dynamics of a loop flanking the active site (Supplementary Fig. 6). This could help reposition the substrate in a more optimal orientation with respect to Arg72 and Phe273 by enhancing enzyme-substrate complementarity (Fig. 6b). Thus, the three-way interaction enables efficient hydrolysis of the organophosphate substrate and constitutes a key intramolecular network that underpins the organophosphatase activity in MPH.

## Discussion

In this study, we used 41 combinatorial landscapes, across seven different enzymes, to quantify apparent idiosyncrasy in adaptive mutations. If one typically expects new mutations, selected over the course of evolution, to be highly context-dependent, we tend to forget that the intramolecular interactions that they create or disrupt within an enzyme may show the same context-dependence. This has strong evolutionary ramifications: Sailer & Harms showed that higher-order epistasis profoundly affects the accessibility of evolutionary trajectories[8]. Therefore, the high prevalence of apparent idiosyncrasy uncovered in our study

builds on previous work by providing the probability (in the form of spread, sign heterogeneity, and deviation from mean effect) of a mutation eliciting an apparently idiosyncratic effect in any given genotype. Furthermore, by specifically focusing our exploration on adaptive landscapes, we paint a very different picture of the prevalence of epistasis in enzymes than previous studies, which found limited specific epistasis among random, non-adaptive mutations[16–18].

The reduced dataset, intended to eliminate cross-landscape correlations, showed a different distribution of sign heterogeneity in SMEs. Our strategy for generating the reduced dataset consisted of retaining landscapes that were more reflective of the adaptive conditions, i.e., probing substrates and metals that were present in the original selection conditions. This resulted in the apparent enrichment of positive and neutral-positive SMEs, and in turn reduced the abundance of positive-negative SMEs (Supplementary Tables 3 and 4). Interestingly, because the SMEs in the reduced dataset were still highly heterogenous (Supplementary Tables 1 and 2), this dataset can be characterized by a stronger presence of magnitude, as opposed to sign, epistasis. Although several studies have explored the differences in patterns of epistasis within singular enzymes across different selection conditions[20,31,34], further research is encouraged to provide a rationale for these differences and to better capture global trends across multiple enzymes.

In agreement with previous studies highlighting the poor predictive power of the biochemical model at the organismal level[8,10], we demonstrate that wt-derived EEs are also poor coefficients for functional predictions along adaptive enzyme trajectories. We acknowledge that there are several advanced prediction models for epistatic analysis utilizing gaussian process regressions[36], classifiers[37], and autoregressive generative models[38]. Indeed, the relative success of these models poses an apparent conundrum: how can the use of sparse epistatic terms achieve successful predictions if pervasive epistasis is at play in adaptive trajectories? We believe that these observations may be reconciled by outlining the key differences between epistatic terms generated by the biochemical model compared to the predictive- or generative- models. The specific interactions derived from genotypes in the biochemical model are highly context-dependent due to pervasive higher-order epistasis, as demonstrated here. The strength underlying predictive and generative models lies in amalgamating each of these specific interactions into a sparser subset of coefficients, which captures global trends within the dataset. However, the terms from these models are ultimately latent variables which, although suitable for prediction and generation of highly-active enzyme variants, are ill-suited for inferring molecular mechanisms, unravelling the topology of intramolecular residue networks, and revealing the extent of apparent idiosyncrasy in adaptive fitness landscapes. We found that employing the biochemical model remains necessary, if we wish to unveil the presence of complex intramolecular networks and evaluate their relevance to understanding the biophysical features that underlie epistasis.

We demonstrated that a large proportion of mutational steps was classified by changes in EEs. These changes were predominantly disruptive, suggesting that throughout evolution, epistatic networks are highly sensitive to the introduction of subsequent adaptive mutations, and can be readily impacted. In fact, diminishing changes in EEs may be connected to another observation in this study, where the use of lower-order idiosyncratic EEs yielded better functional prediction for high-order mutants (Fig. 4b), relative to higher-order EEs. In other words, the existence of alternating fluctuations between positive and negative epistasis in adjacent genotypes, a characteristic of disruptive changes in EEs, will cause over- or under-predictions of function in models operating under the assumption that previously characterized epistasis remains constant at higher orders. Furthermore, the abundance of disruptive changes may provide a molecular justification for the theoretical considerations of idiosyncratic epistasis discussed by

Lyons et al. and Reddy & Desai, where idiosyncratic epistasis resulted in diminishing returns epistasis, as well as increasing costs epistasis[9,39].

We have shown how, in evolutionarily connected genotypes, patterns of epistasis can reveal cryptic functional determinants hidden within intramolecular networks. Our approach aligns with several structural and computational techniques that have been developed over the years to probe amino acid networks[40]. Indeed, our epistasis analysis could be complemented by evolutionary covariation data, which identifies residues that are energetically coupled and likely to co-evolve. Originally supported by statistical coupling analysis (SCA), and later expanded upon by direct coupling analysis (DCA), these statistical frameworks have been instrumental in identifying functionally significant networks, or sectors, in proteins[41–43]. Sectors can be construed as local, intramolecular, subnetworks of residues that support complex communication pathways within a protein structure[44]. These pathways are cardinal to achieve: (i) allostery, via the transduction of a signal from an effector site to a ligand binding site, leading to a reversible functional change[45], or (ii) conformational dynamics that are essential to a protein function[46,47]. Since sectors reflect evolutionarily conserved wiring mechanisms, mutations that alter the regulation and function of these intramolecular networks can rapidly give rise to epistasis. Interestingly, our analysis revealed a high degree of rewiring (Fig. 5) and, for PTE and MPH, evidence of mutational interaction networks that connect through the substrate (Fig. 6). In these unique cases, the ligand constitutes an integral part of the intramolecular network, akin to an amino acid residue, which can facilitate the emergent epistasis. Given limited structural information, with only a handful of mutants' structures being solved, the molecular mechanisms described here remain speculative and must be further probed empirically to ensure their validity. Nevertheless, our mechanistic analysis strongly emphasizes the relevance of exploring network rewiring, evidenced by changes in epistasis, to decipher the molecular mechanisms giving rise to new functions.

Finally, our analysis can be implemented to (i) determine which interactions should be further examined, using downstream structural and mechanistic experiments and (ii) highlight critical interactions that may be incorporated as 'weights' into sequence-based computational models that are typically devoid of structural information. Indeed, phenotypic prediction and intramolecular network analysis have much to benefit from various empirical techniques that can explore, in greater detail, the molecular and structural bases of these networks[40,48]. For instance, molecular dynamics (MD)[46,47] and nuclear magnetic resonance (NMR)[49] could be of great interest to study potential interactions mediated through ligands and co-factors[1]. Likewise, an intramolecular network perspective may help pinpoint the critical elements of a successful active site architecture and highlight the importance of the newly evolved interactions between mutations in 3D space. This contrasts modern sequence-based prediction methods that rely on the specific identity of the mutated residues, as it may allow for the extrapolation of mutations at previously unexplored sites. By combining structural and epistatic information, we may be able to account for network rewiring before engineering campaigns, and to create predictive models rooted in mechanistic information.

## Methods

### PTE combinatorial landscape

During the directed evolution of PTE toward arylesterase activity, we previously identified a cluster of six function-altering mutations[3,35,50]. Here, we constructed one of the combinatorial landscapes analyzed in this study. We explored these six positions on the genetic background of WT PTE (64 variants) and tested all their combinations for activity against 2-naphthyl hexanoate (2NH)[27]. The 64 variants were constructed by site-directed mutagenesis and subcloned into a pET-27-STREP vector[27]. The variants were then transformed into *E. coli* BL21(DE3) carrying the pGro7 plasmid (Takara, Shiga, Japan) for GroEL/ES chaperones

co-expression[50]. Variants were individually inoculated in 96-deep well plates containing lysogeny broth (LB) media, 100 µg/mL ampicillin, and 34 µg/mL chloramphenicol, then grown overnight at 30 °C. Overnight cultures were transferred to a new deep well plate containing LB, supplemented with 100 µg/mL ampicillin, 34 µg/mL chloramphenicol, 200 µM ZnCl$_2$, and 0.2% (w/v) arabinose for chaperone co-expression, then induced with 1 mM IPTG. Pellets were lysed with lysis buffer (50 mM Tris-HCl buffer, 100 mM NaCl, pH 7.5, 0.1% (w/v) Triton-X100, 200 µM ZnCl$_2$, 100 µg/mL lysozyme and 1 µL benzonase (25 U/µL) per 100 mL of lysis buffer). Lysates were incubated with 200 µM 2NH + 1 mM Fast Red and the initial rate of hydrolysis was monitored at 500 nm.

### Data processing

The values of reported functions were divided by the wt background function, or, in the presence of replicates, by the mean of the wt background functions, then log$_{10}$ transformed. However, we raised 10 to the power of all the TEM-1 growth rate values from Mira et al. before using them in our standard pipeline[34], as the growth rates from this study are assumed to be additive and not multiplicative. All processed files are provided (Supplementary Data 1). The non-linear transformation was performed using a four-parameter fit:

$$F = U - \frac{U - L}{1 + e^{(m - F_{add})s}} \tag{1}$$

Where $F$ is the observed function, $F_{add}$ is the predicted function based on a first-order background averaged model (see below), $U$ is the upper bound, $L$ is the lower bound, $m$ is the mid-point, and $s$ is the slope (Supplementary Data 3). Non-linear transformations were applied to datasets wherein the four-parameter model was more parsimonious than linear model, as deemed by the Akaike Information Criterion (AIC). We found that the non-linear transformation corrected fold-change values for the AMC, AMP, CAZ, and TZP trajectories of TEM-1 to binary values, i.e., equal to $U$ or $L$, thus, we removed these landscapes from further analysis. We also removed potentially correlated landscapes, i.e., those probing different measurements, environments, or substrates for the same mutations within an enzyme, producing a reduced data, aimed at eliminating statistical artefacts. To this end, we retained: AP, DHFR c57 trajectory from Lozovsky et al., DHFR Arg trajectory from Tamer et al., MPH in zinc environment, NfsA trajectory 20_39, OXA-48 trajectories 1–3 for ceftazidime, PTE for 2NH, TEM-1 with AM from Mira et al., and TEM-1 from Weinreich et al.

### SME and EE calculation

Genotypes were represented by a string of amino acids that underwent mutation, then encoded using '0' for ancestral states, and '1' for derived states at the given amino acid positions. The single mutational effect (SME) of a mutation at position $i$ was calculated for each genetic background by computing the difference between log-transformed $F_{i=0}$ and $F_{i=1}$. The mutational transition for the SME is denoted with 'x' which represents a transition from 0 to 1 – e.g., the log$_{10}$SME$_{x001}$ represents the SME of mutating position 1 in the position 4 mutant background and is equal to log$_{10} F_{1001}$ – log$_{10} F_{0001}$. Epistatic effects (EEs) between two positions $i$ and $j$ were calculated for each genetic background by computing the difference between log$_{10}$ SME$_{i=x,j=0}$ and log$_{10}$ SME$_{i=x,j=1}$. For example, the log$_{10}$ EE$_{xx00}$ represents the epistatic effect stemming from the interaction between the first and second position mutations in the wt background and is equal to log$_{10}$ SME$_{x100}$ – log$_{10}$ SME$_{x000}$. Importantly, this is equivalent to log$_{10}$ SME$_{1x00}$ – log$_{10}$ SME$_{0x00}$, as the equation can be broken down:

$$\log_{10} EE_{xx00} = \log_{10} SME_{x100} - \log_{10} SME_{x000}$$
$$= (\log_{10} F_{1100} - \log_{10} F_{1000}) - (\log_{10} F_{0100} - \log_{10} F_{0000}) \tag{2}$$

As with pairwise interactions, higher-order IEs were calculated by taking the difference between EEs of the previous order, *e.g.*, $EE_{xxx0} = EE_{xx10} - EE_{xx00}$. This can also be broken down to SME or even $F$:

$$
\begin{aligned}
\log_{10} EE_{xxx0} &= \log_{10} EE_{xx10} - \log_{10} EE_{xx00} \\
&= (\log_{10} SME_{x110} - \log_{10} SME_{x010}) - (\log_{10} SME_{x100} - \log_{10} SME_{x000}) \\
&= ((\log_{10} F_{1110} - \log_{10} F_{0110}) - (\log_{10} F_{1010} - \log_{10} F_{0010})) \\
&\quad - ((\log_{10} F_{1100} - \log_{10} F_{0100}) - (\log_{10} F_{1000} - \log_{10} F_{0000})) \\
&= \log_{10} F_{1110} - \log_{10} F_{1100} - \log_{10} F_{1010} - \log_{10} F_{0110} + \log_{10} F_{0100} \\
&\quad + \log_{10} F_{1000} + \log_{10} F_{0010}
\end{aligned}
$$

$$(3)$$

### Biochemical model

The function of a genotype was predicted using the biochemical model as a sum of all mutational (SME) and epistatic (EE) effects in the wild-type background:

$$
\log_{10} F = \sum_{i=1}^{n} \log_{10} SME_i x_i + \sum_{i<j}^{n} \log_{10} EE_{ij} x_i x_j + \dots \tag{4}
$$

Where $i$ and $j$ represent the index of the residue's position, $x_i$ is either '0' or '1' depending on the mutational state of the residue in the given genotype, and $F$ is the function. This is analogous to the biochemical view of epistasis from Poelwijk et al. [22], however, our model uses $F$ values that do not represent $\Delta G$ of the protein or enzymatic reaction.

### Background-averaged model

Mutations at $n$ residue positions were annotated with variables '-1' for the ancestral state or '1' for the derived state. These were used as $x$ variables in the linear model, $F$ is the log-transformed and wild-type normalized function of the variant. The linear model was constructed such that:

$$
\log_{10} F = \beta_0 + \sum_{i=1}^{n} \beta_i x_i + \sum_{i<j}^{n} \beta_{ij} x_i x_j + \dots + \epsilon \tag{5}
$$

Where $i$ and $j$ represent the index of residue's position, $x_i$ is either '-1' or '1' depending on the mutational state of the residue in the given genotype, $\beta$ are the linear coefficients that represent $SME_{avg}$ and $EE_{avg}$, and $\epsilon$ is the error.

### Reporting summary

Further information on research design is available in the Nature Portfolio Reporting Summary linked to this article.

## Data availability

All raw data can be found in the corresponding references outlined in Table 1. Data for all combinatorial landscapes are provided in $\log_{10}$-transformed wt-normalized format (Supplementary Data 2). Processed data for functional contributions (Supplementary Data 4) and epistasis (Supplementary Data 6) are also available. The structural data for PTE-R1, PTE-R2, PTE-R8, PTE-R18, MPH-Anc, and MPH* are available under PDB accession numbers 4XAF, 4XD5, 4XAY, 4E3T, 6C2C, and 1P9E, respectively. These data are sufficient for the reproduction of all results presented in our work. Data presented in all figures can be found in the Source Data file. Source data are provided with this paper.

## Code availability

Scripts for individual combinatorial landscape analysis and the global statistical analysis are publicly available on GitHub (https://doi.org/10.5281/zenodo.10202238)[51]. The scripts utilize the R language version 4.1.2 (https://www.R-project.org/), along with R packages outlined on GitHub.

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

## Acknowledgements

We thank D. Anderson and the members of the Tokuriki lab for their invaluable comments. This work was supported by the Natural Sciences and Engineering Research Council of Canada (NSERC) Discovery Grant (RGPIN 2017-04909) and the Human Frontier Science Program (HFSP) Research Grant (RGP0054/2020) awarded to N.T.

## Author contributions

K.B. and N.T. conceived and designed the study. K.B performed the statistical analyses. C.M.M. performed the activity assays for the PTE combinatorial landscape. K.B. and C.M.M. performed the molecular mechanism analysis. All authors contributed to writing the paper.

## Competing interests

The authors declare no competing interests.
