## [Peer Review File · Nature Communications]

Pervasive epistasis exposes intramolecular networks in adaptive enzyme evolutionREVIEWER COMMENTS

Reviewer #1 (Remarks to the Author):

In this paper, the authors characterize idiosyncratic epistasis in 45 adaptive landscapes of 7 enzymes. The main claim is that idiosyncratic epistasis is pervasive in enzyme evolution. The authors augment their statistical findings with a discussion of plausible mechanistic scenarios that could explain why mutational and epistatic effects are context-dependent in certain special cases.

Overall, the paper presents several new creative analyses of existing datasets. Unfortunately, these analyses, as presented, do not seem to provide many new insights.

My main concern is that it is not clear what the authors mean by “idiosyncratic” epistasis. They seem to define mutational and epistatic effects as “idiosyncratic” if they exhibit “unique effects in distinct backgrounds” (ll. 54–55). The problem lies in the word “unique”. When “unique” refers to the effect of a mutation, I understand it to mean that the effect in one genetic background is different than its effect in other genetic backgrounds. But this is simply the definition of epistasis (without the qualifier “idiosyncratic”). What makes such epistasis “idiosyncratic”? Similarly, when the word “unique” refers to epistasis between two or more mutations, I understand it to mean that the epistasis coefficients are different in different genetic backgrounds. But this is the standard definition of “higher-order epistasis”. What makes it idiosyncratic?

My understanding of “idiosyncratic” is that it stands in contrast to “global” epistasis (it seems the authors prefer to use “non-specific” instead of “global”, which is fine, as long as we talk about the same thing). By “global” epistasis I mean that the effect of a mutation in a certain background can be predicted using some low-dimensional trait value of the background type (rather than the individual mutations that the background type has). A good example of global epistasis is shown in Figure 4 in Sarkisyan et al (Nature 2016). This “global” predictor may be imperfect, i.e., it may not explain all of the variance in the effects of a mutation across backgrounds above and beyond measurement noise. This means that the effect of the mutation depends on the genetic background not only through the global predictor but also some other (unknown) features. It is such background dependence of mutational effects above and beyond global epistasis that is typically referred to as “idiosyncratic” epistasis (see, for example, Reddy & Desai 2021).

If there is no substance to the qualifier “idiosyncratic”, then it should be dropped from the title and everywhere else in the paper. Then the paper’s main claim is that there is a lot of epistasis in enzyme evolution, which is nothing new. It is of course possible that I misunderstand something, in which case the authors will need to clearly explain what they mean by idiosyncratic. Alternatively, they could adopt the definition of “idiosyncratic” that is already present in the literature (see above) and redo their analyses after “subtracting” global epistasis from their data and analyze the residuals.

OTHER MAJOR ISSUES

1. Another major concern that I have is that not all 45 landscapes are independent, so that the reported statistics may not actually adequately reflect the frequencies of various types of epistasis in different enzymes. For example, SME for the DHFR landscapes from Lozovsky et al all look similar in the presence of different compounds (Supp File 2). I would ask the authors to report how their statistics change if only one landscape per enzyme is used. Alternatively, they could compute correlations between different landscapes for the same enzyme and retain only those that are not correlated.

2. I don't quite understand the point of the first half of section "Idiosyncrasy confounds prediction". The biochemical and the background-averaged models are essentially the same linear model with the only difference being in how the regression coefficients are estimated. In the biochemical model, the coefficients are estimated from fewer data points (only from data in the wildtype background) than in the background-averaged model, so it is obvious that the biochemical model performs worse than the background-averaged model. Moreover, the authors use only the "endpoint" (the most derived) genotype to compare the two models, which also favors the background-averaged model. All in all, I don't see what we learn from this comparison.

3. Likewise, I've also struggled to understand the main point of section "Detecting the change in epistasis from idiosyncratic effects". Here, the authors compare the $(n+1)$ th order epistasis with the n th order epistasis. They classify each n to $n+1$ transition into one of four classes and find that in the majority of cases the sign of epistasis at the higher order is the opposite to the sign of epistasis at the lower order. This is a curious observation. The authors interpret such interactions as "diminishing" and in the discussion they say "diminishing changes provide a molecular justification for the theoretical considerations of idiosyncratic epistasis discussed by Lyons et al. and Reddy & Desai, where idiosyncratic epistasis resulted in diminishing returns epistasis, as well as increasing costs epistasis". This statement seem to imply that the observations presented in this section are consistent with the global epistasis theories by Lyons et al or Reddy & Desai. But if this is true, there must be global epistasis in the data they analyzed, i.e., the phenotype of the background must to some extent predict the effect of the next mutation in this background. Figure 5 indeed gives a hint of a suggestion in this direction. But the authors haven't done the regression analysis to actually demonstrate this. In fact, in the beginning of this section (LL. 296–300), they explicitly rejected the idea of looking for global epistasis in their data, which I found perplexing and unjustified (more on that below). All in all, the analysis in this section seems to be incomplete, and, as a result, the conclusions are unclear.

4. Finally, I found the argument presented in Supplementary File 3 (and referred to in LL. 206–300) inadequate. First of all, the presentation of the argument is frankly substandard, even for a supplementary file. It looks more like a excerpt from a log, not a text that is meant to be read by an uninitiated reader, e.g. text is interspersed with code, figures don't have legends, etc. Second, if I understood it correctly, the basic idea of the argument is the following. The authors made a fake dataset without global epistasis (i.e., where only idiosyncratic epistasis is present), ran one non-linear transformation technique (by Sailer & Harms) that is designed to subtract global epistasis and found that in fact this transformation reduced the original idiosyncratic epistasis. The authors interpret the fact that the original epistasis was reduced as a failure of the Sailer & Harms method. Due to this apparent failure, the authors decided not to use this or other techniques for eliminating global epistasis. But the question

here is not whether this or other similar methods reduce the true epistasis in the absence of global epistasis because in real data we never know whether global epistasis is truly absent. Rather, the question is which model, with or without global epistasis, is a better and more parsimonious explanation for the data. In my opinion, a model where some statistically significant fraction of epistasis is attributed to a non-linear mapping from genotype to phenotype (i.e., a model with global epistasis) is more parsimonious than a model where all of this epistasis is attributed to idiosyncratic effects. If the authors disagree with this, they should present a clear argument why model without global epistasis should be favored over models with global epistasis.

MINOR COMMENTS

L. 165. σ is not defined. In the physics literature, sigma usually refers to standard deviation, but this is not an accepted convention in the biological literature. I suggest the authors replace σ with “standard deviation”, here and throughout the text.

Figure 2a. I don't understand the x axis in Figure 2a. According to methods (LL. 474–475), $SME = F_{\{i=1\}} - F_{\{i=0\}}$, which can take negative values. How do the authors take \log_{10} of negative values?

Figure 2c. Is my understanding correct that the negative-positive section of the pie chart includes all positions where the effect is positive in at least one background and negative in at least one background. In other words, it is possible that 99 backgrounds support one sign (say “+”) and 1 background supports another sign (say, “-“), and such a position would be classified as “negative-positive” even though it is really overwhelmingly positive. To address this issue, would it be possible to visualize and/or quantify what fraction of backgrounds support each mutational sign? Same question for Figure 3b.

I could not find Supplementary Table 5.

Figure 4b. What is “Predicted mutant order” in the figure's x axis?

LL. 319: Where does the percentage 37.7% come from? My understanding is that this is 22.2% + 15.5%. But why do the 18.1% of “No epistasis” cases not count towards “new mutations are often uncoupled from pre-existing mutations”? And why do 22.2% contribute to this?

Figure 5. Would it be possible to plot $EE_{\{xx00\}}$ on the x axis versus $EE_{\{xxyy\}}$ (where $yy = 01, 10$) on the y axis to see more directly how EE changes after an introduction of the next mutation? I would expect a cloud of points mostly below the diagonal. I find this kind of direct comparison of epistasis at the same order across different backgrounds more intuitive to understand than comparing epistasis of different orders.

Reviewer #2 (Remarks to the Author):

Review of NCOMMS-23-24512, Buda et al.
“Pervasive idiosyncratic epistasis...”

This paper addresses the interesting problem of assessing the extent of epistasis along adaptive trajectories in proteins. The motivation is that while other works have tried to analyze the pattern and extent of systematic (reproducible) epistasis through background averaging methods or by probing general epistasis through random mutagenesis, no one has looked carefully at the epistasis along adaptive trajectories (that are obviously relevant for evolving systems). The analysis in this work is driven by a curated selection of published experimental evolution data in multiple protein systems. The curation isolates datasets with combinatorially complete groups of > 4 mutations and collected along adaptive trajectories with selection for a specific phenotype that promotes fitness. With this, the authors claim (1) that there is considerable context dependence (which they call “idiosyncrasy”) of the phenotype of observed single mutations on secondary mutations, (2) that there is considerable context dependence of pairwise epistatic effects (again, called “idiosyncrasy”) on additional mutations, indicating higher-order epistasis, (3) that these epistatic effects profoundly limits the predictive power of biochemical models that only use first and second-order epistatic terms with the wild-type sequence as the reference (so-called single-reference epistasis), (5) that a linear-regression model using background-averaged epistatic terms does a better job at predicting phenotypes of high-order mutants, (6) that a targeted examination of structures of a couple proteins provides more intuitive mechanistic examples of how higher-order epistasis might emerge through the local networks of amino acid interactions. From all this, the authors conclude that epistasis is pervasive along adaptive trajectories and speculate that such epistatic networks could be the substrate for ligand-mediated propagated functional effects within proteins.

In general, this is a very nice paper. The topic is a central one in understanding the relationship of genotype and phenotype in evolving systems, the analysis is intelligently executed, and the specific conclusions seem well-supported by the data analysis. More generally, the concept of looking at epistasis along adaptive trajectories under selection rather than through random mutagenesis or by averaging epistatic terms over an ensemble of solutions is fundamentally interesting and clarifying. The work will add to the important discussion of the architecture of epistasis in protein systems. Some specific points the authors should consider:

(1) The central (correct) idea behind this work is context-dependence of mutational effects along adaptive trajectories, and the need to be able to know at least some of the key epistatic terms to be able to predict the allowed trajectories during evolution under selection. The mutational paths are not random and indeed, are highly likely to be funneled into networks of amino acid positions that are capable of contributing adaptive mutations. The study in this work (while admirable) is but a very sparse sampling of possible amino acid mutations that can comprise the full ensemble of potential trajectories. More importantly, potential intramolecular architectures that might constrain and simplify the adaptive landscape will be invisible to this scale of analysis. For these reasons, this reviewer would strongly urge the authors to reconsider the use of the term “idiosyncrasy” in order to future-proof their paper. After all, the fundamental idea here is about context-dependence caused by the presence of high-order

epistasis (to an unknown extent) in adaptive positions. Whether this is true idiosyncrasy within the protein structure or variation within a constrained and ultimately know-able architecture that masquerades as such given the limited sampling of data here is something to be seen.

(2) The notion of intramolecular networks that both mediate active-site dependent allosteric signal transmission and provide for adaptive mutations has been presented in the literature. It might be valuable for the authors to discuss this body of work (which not inconsistent with the present study) in the context of their findings here.

(3) Recent studies (and the data presented here) support the predictive power of background-averaged forms of epistasis to predict phenotypes of complex mutations with just a few critical parameters (albeit including some higher-order terms). Similarly, sequence-based computational models (which represent a kind of background averaging over homologs) have provided generative models that can design artificial proteins with a relatively small number of epistatic terms. Could the authors discuss the claims in this work in the context of these findings? In some sense, reconciling the generative power of sparse epistasis models with the findings of dense context-dependence along specific evolutionary trajectories goes to the heart of the problem. Much like random-walk diffusion, there is a big difference between the information required to know statistical outcomes in evolution versus the reconstruction of specific paths.

1. Revisions requested by Reviewer #1:

- **Major Revision #1:**

My main concern is that it is not clear what the authors mean by “idiosyncratic” epistasis. They seem to define mutational and epistatic effects as “idiosyncratic” if they exhibit “unique effects in distinct backgrounds” (ll. 54–55). The problem lies in the word “unique”. When “unique” refers to the effect of a mutation, I understand it to mean that the effect in one genetic background is different than its effect in other genetic backgrounds. But this is simply the definition of epistasis (without the qualifier “idiosyncratic”). What makes such epistasis “idiosyncratic”? Similarly, when the word “unique” refers to epistasis between two or more mutations, I understand it to mean that the epistasis coefficients are different in different genetic backgrounds. But this is the standard definition of “higher-order epistasis”. What makes it idiosyncratic? My understanding of “idiosyncratic” is that it stands in contrast to “global” epistasis (it seems the authors prefer to use “non-specific” instead of “global”, which is fine, as long as we talk about the same thing). By “global” epistasis I mean that the effect of a mutation in a certain background can be predicted using some low-dimensional trait value of the background type (rather than the individual mutations that the background type has). A good example of global epistasis is shown in Figure 4 in Sarkisyan et al (Nature 2016). This “global” predictor may be imperfect, i.e., it may not explain all of the variance in the effects of a mutation across backgrounds above and beyond measurement noise. This means that the effect of the mutation depends on the genetic background not only through the global predictor but also some other (unknown) features. It is such background dependence of mutational effects above and beyond global epistasis that is typically referred to as “idiosyncratic” epistasis (see, for example, Reddy & Desai 2021). If there is no substance to the qualifier “idiosyncratic”, then it should be dropped from the title and everywhere else in the paper.

Response to MR #1a:

We thank Reviewer 1 for addressing this issue – indeed, the definition of ‘idiosyncrasy’ varies across the literature {PMIDs: 37226182, 33634468, 30037990, 32895516}. As such, we have modified our definition of ‘idiosyncratic’ to emulate the one proposed by Lyons *et al.* {PMID: 32895516}: “idiosyncratic (*i.e.*, varying with the interacting nucleotides involved)”, and have defined it clearly in LL. 54–55. Though the variability of effects in different backgrounds is caused by epistasis, we are interested in the extent of this variability. While epistasis and higher-order epistasis are the cause of this apparent idiosyncrasy, the purpose of our analysis is to survey the extent of idiosyncrasy in SMEs and EEs.

Furthermore, after considering comment #13 from Reviewer 2, we acknowledge that the term ‘idiosyncratic’ can be ambiguous: in most cases, this terminology is used in reference to a fitness landscape, whereby an SME or EE appears idiosyncratic within the given set of tested combinations (in contrast to ‘global’, as addressed by the reviewer). Indeed, apparent idiosyncrasies can disappear after performing global transforms or partitioning the variance into several, landscape-wide coefficients. In reality, a full description of the idiosyncrasy of SMEs and EEs in a protein will always be out of reach: it would require the sampling of the entire theoretical fitness landscape of a protein. This suggests that any uncovered idiosyncrasies can only be ‘apparent’ since they reflect an idiosyncratic behavior relative to a specific, and inherently limited, reference dataset.

Thus, we have made two major revisions to this manuscript. First, we have modified all instances of the term ‘idiosyncratic’ throughout our manuscript to reflect the ‘apparent’ nature of these effects (akin to ‘apparent’ rate/binding constants in enzymology, *e.g.*, K_{app} , given the limited experimental regime used to extract them) in the given dataset. We also modified our title accordingly: “*Pervasive epistasis*

exposes intramolecular networks in adaptive enzyme evolution". Second, as idiosyncrasy is a qualifier, it cannot be estimated explicitly as formerly described in LL. 171–234 and 236–279. Thus, we have renamed the quantitative metric to 'heterogeneity', characterized by the spread, sign-variability, and wt context-dependence. We hope that these changes clarify and emphasize an apparent context-dependence for every adaptive effect, stemming from epistasis, which can distort predictions when unaccounted for by appropriate models (Fig. 4), but can help reveal unique changes in specific genotypes along adaptive evolutionary trajectories (Figs. 5 and 6).

Then the paper's main claim is that there is a lot of epistasis in enzyme evolution, which is nothing new. It is of course possible that I misunderstand something, in which case the authors will need to clearly explain what they mean by idiosyncratic. Alternatively, they could adopt the definition of "idiosyncratic" that is already present in the literature (see above) and redo their analyses after "subtracting" global epistasis from their data and analyze the residuals.

Response to MR #1b:

We greatly appreciate the suggestion of probing our dataset after performing a non-linear transformation. We have followed the reviewer's advice by implementing a four-parameter non-linear transformation on all fitness landscapes. Notwithstanding this new analysis, we found that all our previous conclusions remain unchanged, which greatly improves the robustness of our analysis. We outline the details of this process in our response to MR #5.

However, we must disagree with the claim that "there is a lot of epistasis in enzyme evolution, which is nothing new". The two main commentaries on the abundance of epistasis in evolutionary trajectories by Weinreich *et al.* and Sailer & Harms highlight significant levels of epistasis (including higher-order epistasis), however, they do so across a combination of protein- and genomic-level fitness landscapes {PMIDs: 24290990, 28505183}. We discuss this in LL. 64–68. The protein-centric view, on the other hand, as outlined in the influential review entitled '*Epistasis in Protein Evolution*' by Starr & Thornton {PMID: 26833806}, highlights "[the lack of] a consensus view" on the matter. They report that some studies present epistasis as "rampant {PMID: 20975933}, whereas others claim that the frequency and magnitude of epistasis are sufficiently low such that it does not strongly affect the patterns of substitution in evolving proteins {PMID: 26226986}". Several surveys conducted with deep mutational scanning (DMS) report that pairwise epistatic effects greater than 2-fold only represent 5% of all combinations in protein G domain 1 {PMID: 25455030} and ~14% in PSD95 PDZ3 domain {PMID: 33620761}. Furthermore, studies that do highlight the prevalence of epistasis in proteins often provide metrics that demonstrate that most of the phenotypic variability can be recapitulated by non-linear mapping or simple pairwise interactions {PMIDs: 34162839, 37732229}. In our manuscript, we note that all these studies were conducted on landscapes that include random, non-adaptive mutations (LL. 71–74). **These observations may lead the field to believe that: (i) although epistasis is prevalent at the genomic level, it is generally sparse in proteins and that (ii) most of the phenotypic variability in proteins can be effectively captured by non-linear transformations and pairwise effects.**

Our study contrasts this body of work in two novel ways. First, **we specifically focus on adaptive evolutionary landscapes of enzymes** and avoid probing random, non-selective mutations on non-enzymatic targets (*e.g.*, by DMS). We make this clear by using the term 'adaptive' throughout the manuscript (LL. 72, 75, 103, 184, 230–232, 305, 342, 359, 400, 442, 456, 465). We have also further

emphasized this by modifying the title to contain the term ‘adaptive enzyme evolution’. Second, **we attempt to provide metrics for the perceived level of ‘randomness’ in mutational and epistatic effects for experimentally relevant reference genotypes** (which we refer to as “apparent idiosyncrasy”). We believe that these contributions will be invaluable to the protein evolution and engineering communities, as they are more relevant to their pursuits, and provide more intuitive metrics for their purposes. To emphasize all of these points, we have made corresponding changes in the abstract (LL. 17–18), introduction (LL. 70–71), and discussion (LL. 447-453).

- **Major Revision #2:**

Another major concern that I have is that not all 45 landscapes are independent, so that the reported statistics may not actually adequately reflect the frequencies of various types of epistasis in different enzymes. For example, SME for the DHFR landscapes from Lozovsky et al all look similar in the presence of different compounds (Supp File 2). I would ask the authors to report how their statistics change if only one landscape per enzyme is used. Alternatively, they could compute correlations between different landscapes for the same enzyme and retain only those that are not correlated.

Response to MR #2:

This is an excellent point. Indeed, while several of the studies analyzed here have found key differences in epistatic patterns for the same set of mutations across different substrates and environments {PMID: 34162839, 25946134, 32745183}, there is, in fact, a risk of including pseudo-replicates originating from a correlation between specific genotype-environment combinations. To this end, we repeated our analysis (as suggested) and only retained landscapes with unique mutations, *i.e.*, we focused on studies that probed the same enzyme, as long as the studied mutations differed in identity. We briefly address this in LL. 168-169 and elaborate on which landscapes were retained in LL. 550-555. We found that only one of our metrics was significantly affected by removing correlated landscapes: the proportion of positive-negative sign contribution for positions decreased from 47% to 23% (LL. 217-219 and 231). Though this suggests that sign variability for SMEs may be inflated by landscape correlations, we found that all other values remain relatively similar, and mention this throughout the text (LL. 208-209, 228-229, 266-268, 303-304, 356-357). Furthermore, we now provide a full list of all statistics for this reduced dataset in the supplementary section (Supplementary Tables 2, 4, 6, 8, 10).

- **Major Revision #3:**

I don’t quite understand the point of the first half of section “Idiosyncrasy confounds prediction”. The biochemical and the background-averaged models are essentially the same linear model with the only difference being in how the regression coefficients are estimated. In the biochemical model, the coefficients are estimated from fewer data points (only from data in the wildtype background) than in the background-averaged model, so it is obvious that the biochemical model performs worse than the background-averaged model. Moreover, the authors use only the “endpoint” (the most derived) genotype to compare the two models, which also favors the background-averaged model. All in all, I don’t see what we learn from this comparison.

Response to MR #3:

We fully agree with the reviewer's evaluation of the background-averaged (BA) model: mathematically, the BA model will always outperform (or, in rare cases, match) the predictive power of the biochemical model. We have included the BA model as a positive control – we simply use it to demonstrate that phenotypic complexity can be statistically captured by relatively few epistatic coefficients in our datasets. We, in fact, highlight this unsurprising outcome in line 294 using the phrase “as expected”. What we found notable, however, is the consistently poor performance of the biochemical model, even at increasing orders. This comparison demonstrates that higher-order epistasis confounds the predictions of the biochemical model and that the apparent idiosyncrasies explored in previous sections are being incorporated into the model, resulting in incorrect phenotypic predictions of further derived genotypes. This is complemented by the second part of the analysis, where instead of generating predictions for the endpoint variant, we demonstrate that the biochemical model can often result in poorer predictions of evolutionary intermediates when given access to more data (*i.e.*, using higher orders of the biochemical model), further emphasizing the role of apparent idiosyncrasy in this matter.

- **Major Revision #4:**

Likewise, I've also struggled to understand the main point of section “Detecting the change in epistasis from idiosyncratic effects”. Here, the authors compare the (n+1)th order epistasis with the nth order epistasis. They classify each n to n+1 transition into one of four classes and find that in the majority of cases the sign of epistasis at the higher order is the opposite to the sign of epistasis at the lower order. This is a curious observation. The authors interpret such interactions as “diminishing” and in the discussion they say “diminishing changes provide a molecular justification for the theoretical considerations of idiosyncratic epistasis discussed by Lyons et al. and Reddy & Desai, where idiosyncratic epistasis resulted in diminishing returns epistasis, as well as increasing costs epistasis”. This statement seem to imply that the observations presented in this section are consistent with the global epistasis theories by Lyons et al or Reddy & Desai. But if this is true, there must be global epistasis in the data they analyzed, *i.e.*, the phenotype of the background must to some extent predict the effect of the next mutation in this background. Figure 5 indeed gives a hint of a suggestion in this direction. But the authors haven't done the regression analysis to actually demonstrate this. In fact, in the beginning of this section (LL. 296–300), they explicitly rejected the idea of looking for global epistasis in their data, which I found perplexing and unjustified (more on that below). All in all, the analysis in this section seems to be incomplete, and, as a result, the conclusions are unclear.

Response to MR #4:

We thank Reviewer 1 for raising these discussion points. The theories of global epistasis patterns presented in Lyons *et al.* and Reddy & Desai are, indeed, commentaries on global epistasis patterns, which our analysis has thus far avoided. As addressed in MR #2 and MR #5, we have performed the network rewiring investigation both on an uncorrelated landscape dataset, as well as on non-linearly transformed data, the latter of which partially addresses the reviewer's point regarding the acknowledgment of global epistasis. Furthermore, we referenced these two studies with the hope of encouraging future research on the connection between the molecular mechanisms underpinning epistasis and their contribution to the genetic architecture of fitness landscapes, but do not want to imply that our analysis is directly connected to the claims in these studies. In particular, the study by Reddy & Desai, later elaborated on by Bakerlee

et al. {PMID: 35511982} in a follow-up study and a review by Johnson *et al.* {PMID: 37226182}, demonstrates how fitness-correlated trends can be dictated by few “idiosyncratic” effects. We believe that the analysis of fitness-correlated trends and the contribution of specific idiosyncrasies on the fitness landscape architecture of our data is beyond the scope of our study. Our exploration of n to $n+1$ transitions is primarily intended to introduce a network-centric representation of proteins and highlight a possible link between network rewiring and epistasis; we elaborate on this point in MR #12. These network rewiring patterns are then utilized to provide novel, structure-based insights into the molecular mechanisms. To emphasize these points, we altered the nomenclature of rewiring patterns to ‘constructive’ and ‘disruptive’ (LL. 354-355 and Fig. 5), in place of ‘enhancing’ and ‘diminishing’, to reflect the network-centric nature of the analysis, as well as eliminating the strong link to the concept of ‘diminishing returns’, which we do not explicitly analyze indeed. We have also modified our discussion section in LL. 463–474 to solely speculate on the link between our analysis and the theoretical considerations of idiosyncratic epistasis.

- **Major Revision #5:**

Finally, I found the argument presented in Supplementary File 3 (and referred to in LL. 206–300) inadequate. First of all, the presentation of the argument is frankly substandard, even for a supplementary file. It looks more like an excerpt from a log, not a text that is meant to be read by an uninitiated reader, e.g. text is interspersed with code, figures don’t have legends, etc. Second, if I understood it correctly, the basic idea of the argument is the following. The authors made a fake dataset without global epistasis (i.e., where only idiosyncratic epistasis is present), ran one non-linear transformation technique (by Sailer & Harms) that is designed to subtract global epistasis and found that in fact this transformation reduced the original idiosyncratic epistasis. The authors interpret the fact that the original epistasis was reduced as a failure of the Sailer & Harms method. Due to this apparent failure, the authors decided not to use this or other techniques for eliminating global epistasis. But the question here is not whether this or other similar methods reduce the true epistasis in the absence of global epistasis because in real data we never know whether global epistasis is truly absent. Rather, the question is which model, with or without global epistasis, is a better and more parsimonious explanation for the data. In my opinion, a model where some statistically significant fraction of epistasis is attributed to a non-linear mapping from genotype to phenotype (i.e., a model with global epistasis) is more parsimonious than a model where all of this epistasis is attributed to idiosyncratic effects. If the authors disagree with this, they should present a clear argument why model without global epistasis should be favored over models with global epistasis.

Response to MR #5:

We appreciate these constructive comments regarding the formatting issues in Supplementary File 3. We have now modified the file (now Supplementary File 2) to be much more legible to all readers. Furthermore, as we have alluded to in the response to MR #1b, the new file now outlines our non-linear transformation process. We tested transformations using (i) monotonic splines and (ii) a four-parameter fit, and assessed the model using an AIC to determine whether the transformation is more parsimonious than a linear model (i.e., no transformation). We have outlined this in the method section (LL. 541-550) and briefly in LL. 166-168. We also found that for four landscapes (see LL. 143-145), the four-parameter fit was more parsimonious than untransformed data, but resulted in a transformation of the phenotype onto a binary scale, thus severely impacting the resolution of mutational effects. Thus, we excluded these

landscapes from our datasets, which in turn reduced the number of analyzed genotype-phenotype measurements. After performing our analysis on the transformed data, we note that some of the reported metrics have changed, however, these changes do not appear to be significant and all of our previous conclusions remain valid. We have corrected all values and figures to reflect these changes in the text. Finally, we have removed the previous justification of not using non-specific epistasis (LL. 331–334).

- **Minor Revision #6:**

L. 165. σ is not defined. In the physics literature, sigma usually refers to standard deviation, but this is not an accepted convention in the biological literature. I suggest the authors replace σ with “standard deviation”, here and throughout the text.

Response to MR #6:

We have changed all instances of 2σ to 2 SD (LL. 180–181, Fig 2, 202, 205, 207–208, Fig 3, and all Supplementary Tables).

- **Minor Revision #7:**

Figure 2a. I don’t understand the x axis in Figure 2a. According to methods (LL. 474–475), $SME = F_{\{i=1\}} - F_{\{i=0\}}$, which can take negative values. How do the authors take \log_{10} of negative values?

Response to MR #7:

The figure legend and methods were, indeed, inconsistent – we thank the reviewer for identifying this error. We have corrected our methods section to highlight that all data are log-transformed (LL. 560-587), thus, \log_{10} SMEs can take on negative values but SMEs cannot.

- **Minor Revision #8:**

Figure 2c. Is my understanding correct that the negative-positive section of the pie chart includes all positions where the effect is positive in at least one background and negative in at least one background. In other words, it is possible that 99 backgrounds support one sign (say “+”) and 1 background supports another sign (say, “-”), and such a position would be classified as “negative-positive” even though it is really overwhelmingly positive. To address this issue, would it be possible to visualize and/or quantify what fraction of backgrounds support each mutational sign? Same question for Figure 3b.

Response to MR #8:

This was an insightful question that we analyzed according to the reviewer’s suggestion: we opted to provide a visual (semi-quantitative) analysis of these proportions across all positions (SMEs) and combinations (EEs). We tested the proportion of both the positive-neutral-negative, as well as only positive-negative proportions after removing all neutral effects (*i.e.*, strictly exploring positives *versus* negatives even in the presence of neutral effects). We found that the proportions were variable, with a substantial representation of positions and combinations that exhibited a 50:50 ratio of positive-negative

effects. We present these data in supplementary figures 2 and 4, as well as in-text comments throughout the text (LL. 215-217 and 255-256).

- **Minor Revision #9:**

I could not find Supplementary Table 5.

Response to MR #9:

We apologize for this oversight. We have added this table (now, due to the additional data, named Supplementary Table 10). We have also included the previously missing Supplementary Table 6 (now appearing as Supplementary Table 11).

- **Minor Revision #10:**

Figure 4b. What is “Predicted mutant order” in the figure’s x axis?

Response to MR #10:

The “predicted mutant order” is the mutational order of the genotype along the most accessible trajectory for each of the landscapes. For example, predicted mutant order 4 a genotype, harbouring 4 mutations relative to the wt, that is found on the most accessible trajectory. We hope that the additional clarification we provide in the legend for Fig. 4 (LL. 324–325) is sufficient.

- **Minor Revision #11:**

LL. 319: Where does the percentage 37.7% come from? My understanding is that this is 22.2% + 15.5%. But why do the 18.1% of “No epistasis” cases not count towards “new mutations are often uncoupled from pre-existing mutations”? And why do 22.2% contribute to this?

Response to MR #11:

We would like to thank the reviewer for spotting this error: mutational uncoupling should, indeed, be a sum of the ‘no new epistasis’ and ‘no epistasis’ categories. The value should have been 18.1% + 15.5% = 33.6%, while now, considering the new data after performing a non-linear transformation, it is 23.6% + 14.3% = 37.9% as seen in line 359 (and 39.5% in the uncorrelated landscape data set; see Supplementary Table 12)

- **Minor Revision #12:**

Figure 5. Would it be possible to plot $EE_{\{xx00\}}$ on the x axis versus $EE_{\{xxyy\}}$ (where $yy = 01, 10$) on the y axis to see more directly how EE changes after an introduction of the next mutation? I would expect a cloud of points mostly below the diagonal. I find this kind of direct comparison of epistasis at the same order across different backgrounds more intuitive to understand than comparing epistasis of different orders.

Response to MR #12:

Though we find this visualization of EE changes interesting, we believe that the descriptive statistics that we provide in Fig. 5 are already well-suited to reflect these trends. A plot of $EE_{\{xx00\}}$ versus $EE_{\{xxyy\}}$ would, indeed, be comparing epistasis of the same order across different backgrounds, however, due to the nature of the biochemical model, the exploration of n to $n+1$ transitions already contains this information. For example, when we consider the transition from $EE_{\{xx00\}}$ to $EE_{\{xxx0\}}$ and find that the EE value changes signs (an example of a disruptive change) we necessarily know that the absolute value of $EE_{\{xx10\}}$ is less than the absolute value of $EE_{\{xx00\}}$ because $EE_{\{xxx0\}} = EE_{\{xx10\}} - EE_{\{xx00\}}$ or $EE_{\{xx10\}} = EE_{\{xxx0\}} + EE_{\{xx00\}}$ (this example is outlined in eq. 3 in LL. 566–569). Furthermore, we do not see a particularly strong bias towards either positive or negative EEs in our dataset (Fig. 2), thus it is reasonable to assume that one would expect points from an EE_{wt} (e.g., $EE_{\{xx00\}}$) versus $EE_{not\ wt}$ (e.g., $EE_{\{xxyy\}}$) to be equally distributed above and below the diagonal, given the high propensity for disruptive over constructive change in epistasis trends (Fig. 5). In fact, the point distribution would likely resemble a flatter-than-diagonal distribution, i.e., negative EE_{wt} would show positive $EE_{not\ wt}$ and positive EE_{wt} would show negative $EE_{not\ wt}$, though several points would not follow this trend as disruptive changes represent 28.6% of all possible EE transitions (37.6% * 76%).

2. Revisions requested by Reviewer #2:

- **Revision #13:**

The central (correct) idea behind this work is context-dependence of mutational effects along adaptive trajectories, and the need to be able to know at least some of the key epistatic terms to be able to predict the allowed trajectories during evolution under selection. The mutational paths are not random and indeed, are highly likely to be funneled into networks of amino acid positions that are capable of contributing adaptive mutations. The study in this work (while admirable) is but a very sparse sampling of possible amino acid mutations that can comprise the full ensemble of potential trajectories. More importantly, potential intramolecular architectures that might constrain and simplify the adaptive landscape will be invisible to this scale of analysis. For these reasons, this reviewer would strongly urge the authors to reconsider the use of the term “idiosyncrasy” in order to future-proof their paper. After all, the fundamental idea here is about context-dependence caused by the presence of high-order epistasis (to an unknown extent) in adaptive positions. Whether this is true idiosyncrasy within the protein structure or variation within a constrained and ultimately know-able architecture that masquerades as such given the limited sampling of data here is something to be seen.

Response to Rev #13:

We thank Reviewer 2 for this constructive feedback, particularly the astute point regarding the limited sampling of data and, therefore, an ever-present uncertainty in defining effects as truly idiosyncratic within the scope of an entire protein landscape. This response, in conjunction with the comments in MR #1a from Reviewer 1, has led us to make several modifications. First, we have modified our definition of ‘idiosyncratic’ to closely follow the one proposed by Lyons *et al.* {PMID: 32895516}, “idiosyncratic (i.e., varying with the interacting nucleotides involved)”, and have defined it clearly in the text in LL. 54–55. Second, we have modified all instances of the term idiosyncratic in our manuscript to reflect the ‘apparent’ nature of these effects (akin to ‘apparent’ rate/binding constants in enzymology, e.g., $K_{i\ app}$, given the limited experimental regime used to extract them) in the given dataset. Finally, as idiosyncrasy is a qualifier, it cannot be estimated explicitly as we have previously done in LL. 171–234 and 236–279,

thus, we have renamed the quantitative metric to ‘heterogeneity’, characterized by the spread, sign-variability, and wt context-dependence. We hope that these changes emphasize an apparent context-dependence for every adaptive effect, without making a claim that these dependencies are “true idiosyncrasies” in the context of an entire protein’s fitness landscape.

- **Revision #14:**

The notion of intramolecular networks that both mediate active-site dependent allosteric signal transmission and provide for adaptive mutations has been presented in the literature. It might be valuable for the authors to discuss this body of work (which not inconsistent with the present study) in the context of their findings here.

Response to Rev #14:

This is an excellent suggestion to provide additional resources for readers to interpret our findings in the context of the rich literature within the field of intramolecular protein networks. We have fully revamped the fourth paragraph of our discussion (LL. 475–503) to include discussions regarding allosteric signal transmissions as well as conformational dynamics in the context of intramolecular networks. We have also provided methods of probing intramolecular networks that complement ours, namely direct coupling analysis (DCA) and protein “sectors” exploration.

- **Revision #15:**

Recent studies (and the data presented here) support the predictive power of background-averaged forms of epistasis to predict phenotypes of complex mutations with just a few critical parameters (albeit including some higher-order terms). Similarly, sequence-based computational models (which represent a kind of background averaging over homologs) have provided generative models that can design artificial proteins with a relatively small number of epistatic terms. Could the authors discuss the claims in this work in the context of these findings? In some sense, reconciling the generative power of sparse epistasis models with the findings of dense context-dependence along specific evolutionary trajectories goes to the heart of the problem. Much like random-walk diffusion, there is a big difference between the information required to know statistical outcomes in evolution versus the reconstruction of specific paths.

Response to Rev #15:

We would like to thank the reviewer for posing this question, as we believe it provides an excellent opportunity to contextualize our study within the field of contemporary epistasis analysis. Indeed, strong context-dependence along specific evolutionary trajectories may require a deeper understanding of mutational effects beyond sequence information, and we speculate in LL. 504–507 that structural information may somehow be employed in a ‘weighting’ regime to aid sequence-based models. Furthermore, we propose that network understanding may be used to construct structure-based models of an ‘optimal’ active site architecture, with the constraint of maintaining previously identified epistatic interactions. Such analyses could propose entirely novel mutations that have not yet been probed *via* sequence-based methods (LL. 511–515).

REVIEWERS' COMMENTS

Reviewer #1 (Remarks to the Author):

The authors have put substantial effort into this revision. In particular, I appreciate that they did the non-linear regression analysis and the fact that they redid their analyses on the reduced set of unrelated landscapes. As a result of this additional work, the claims became more robust. All my comments at this stage are minor and are mainly aimed at improving the presentation of the results. Overall, this is thorough reanalysis of existing datasets that comprehensively characterizes patterns of epistasis among adaptive mutations within proteins. As such, it is a valuable contribution to the field.

MINOR COMMENTS

1. Regarding the section “Apparent idiosyncrasy confounds prediction”, I think I understand now that the authors are trying to demonstrate that the knowledge of effects and interactions in any given genetic background is not sufficient to fully predict the fitnesses of genotypes that are several mutations away. I suppose my confusion stems from the fact that the background-averaged model is not necessary to make this point. It would be sufficient to show only the failure of the biochemical model in Figure 4a. But the authors present a direct comparison of the two models, which forces the reader to think about why one model is worse than the other, but this question is not what the authors are concerned with. It seems that for the clarity of the presentation, the background averaged model can be removed altogether or put in a separate (perhaps supplementary?) figure.
2. I appreciate that the authors changed sigma to SD, but I think they should still define what “SD” means the first time they use this abbreviation.
3. In Figures 2b, 2d, 3a, 3c, I would suggest removing vertical lines corresponding to 2-fold, 5-fold and 10-fold differences. It makes sense to show the 1.5-fold line because this corresponds to the significance threshold, but the other lines do not add much new information and clutter the plots.
4. The fact that the reduced dataset shows a smaller proportion of negative-positive positions than the full dataset is interesting. It would be good if the authors added a comment interpreting this observation.

Reviewer #2 (Remarks to the Author):

Second review of NCOMMS-23-24512, Buda et al.
“Pervasive idiosyncratic epistasis...”

This review addresses a revision of this paper aimed at addressing the extent of epistasis along adaptive trajectories in proteins. The work makes use of published data on experimental evolution to examine combinatorially complete groups of > 4 mutations for the prevalence of epistasis. As before, the central claims are (1) that there is considerable context-dependence of mutational effects, (2) that this property limits the predictive power of biochemical models based on single-reference epistasis, (3) that background-averaged epistatic models do better, and (4) that examination of many specific instances permits hypotheses for the structural basis of these pairwise and higher-order epistatic interactions. The basic conclusion is that epistasis is pervasive along specific adaptive trajectories.

As before, the opinion of this reviewer is that this is a nice paper that adds to our general analysis of epistasis in proteins and its ramifications for mechanism and evolutionary processes. In revision, the authors have addressed most of the issues raised in the initial review, especially with regard to clarifying what is meant by “idiosyncrasy”. As a consequence, the paper is much improved. However, there are a couple of points that should be considered and corrected by the authors. Specifically:

1. One suggestion was to connect this work to a substantial past record of work on intramolecular epistatic networks that contain the high-order interactions of amino acids and the connection of these networks to ligand-mediated signal propagation and allosteric communication. The authors have responded with a new paragraph describing statistical frameworks for analyzing coevolution of amino acids and the emergent concept of protein sectors. The authors should correct the discussion to be clear that the ideas of sectors came from direct statistical decompositions of empirical correlations (statistical coupling analysis (SCA)) and only later formalized in generative Potts models such as DCA. The ideas described by the authors are more or less correct, but the historical record should be made correct.
2. A second (admittedly non-trivial) issue was to provide some rationale to explain an apparent inconsistency between the concept of pervasive epistasis along adaptive trajectories and the finding that background averaged epistasis in proteins can be sparse and yet near-complete with regard to predictive and even generative power. Indeed, how can we get away with a sparse representation that can generate proteins with fitness above a selection threshold if there is extensive epistasis in the paths that link such solutions? In this regard, it is not fully clear that the authors have addressed the central issue. In the opinion of this reviewer, it is not about the need to integrate structural and epistatic information. This is about the fact that adaptive trajectories collected with positive selection will focus adaptive mutations in highly epistatic units within proteins and that within such units, the effects of lower-order mutations will be highly context dependent (due to the existence of sparse but high-order epistatic terms not exposed in these works). Generative models learn approximate representations of these collective epistatic terms and can thus be sparse. The authors need not reconstruct these arguments in this current work, but at the least they can point out the apparent inconsistency as an issue for further work.

1. Revisions requested by Reviewer #1:

- **Minor Revision #1:**

Regarding the section “Apparent idiosyncrasy confounds prediction”, I think I understand now that the authors are trying to demonstrate that the knowledge of effects and interactions in any given genetic background is not sufficient to fully predict the fitnesses of genotypes that are several mutations away. I suppose my confusion stems from the fact that the background-averaged model is not necessary to make this point. It would be sufficient to show only the failure of the biochemical model in Figure 4a. But the authors present a direct comparison of the two models, which forces the reader to think about why one model is worse than the other, but this question is not what the authors are concerned with. It seems that for the clarity of the presentation, the background averaged model can be removed altogether or put in a separate (perhaps supplementary?) figure.

Response to MR #1:

Indeed, as mentioned by the reviewer, the main aim of this analysis was to illustrate how apparent idiosyncrasy distorts predictions when using the biochemical model. By including the background-averaged model, we wanted to demonstrate that the endpoint variants are not intrinsically “unpredictable”, since a 3rd-order background-averaged model was sufficient for accurate prediction. Nevertheless, we agree that the identification of predictable *versus* unpredictable variants within our data is not the focus of our analysis. Consequently, we followed the reviewer’s advice and have moved the background-averaged model to Supplementary Fig. 5, retaining only the biochemical model predictions in the main text (LL. 239–266).

- **Minor Revision #2:**

I appreciate that the authors changed sigma to SD, but I think they should still define what “SD” means the first time they use this abbreviation.

Response to MR #2:

We thank the reviewer for noting this oversight: we have defined the acronym in line 169 and use SD in all following instances of the term. Although we previously defined it in line 158, this was done in parentheses and in reference to the median error rate calculation. We decided to remove this instance of the acronym in order to provide the more explicit definition in line 169.

- **Minor Revision #3:**

In Figures 2b, 2d, 3a, 3c, I would suggest removing vertical lines corresponding to 2-fold, 5-fold and 10-fold differences. It makes sense to show the 1.5-fold line because this corresponds to the significance threshold, but the other lines do not add much new information and clutter the plots.

Response to MR #3:

The 1.5-fold threshold is our significance threshold (based on our data), while the remaining thresholds were included to illustrate how the metrics change when examined under less stringent thresholds. However, as we provide all these data in the supplementary material (Supplementary Tables 1–2 and 5–8; Supplementary Figs. 1b and 3), we have followed the reviewer’s advice and removed these indicators to declutter the plots in the main text.

- **Minor Revision #4:**

The fact that the reduced dataset shows a smaller proportion of negative-positive positions than the full dataset is interesting. It would be good if the authors added a comment interpreting this observation.

Response to MR #4:

This is indeed an interesting observation; we appreciate the reviewer's inquiry into this phenomenon. Aside from the smaller proportion of positive-negative positions, upon consulting all heterogeneity metrics, we also noticed that **1)** the magnitude of heterogeneity was similar between the full and reduced datasets (*i.e.*, \log_{10} 2SD of SMEs), **2)** the deviation of SME_{wt} to SME_{avg} was similar in the two datasets, and **3)** the reduced dataset was enriched for neutral-positive mutations. We reasoned that this may be the consequence of selection bias: to ensure the reduced dataset represented as many adaptive environments as possible, we preferentially removed fitness landscapes that probed adaptive mutations under "non-adaptive" selection conditions, *i.e.*, ones not used in the directed evolution experiments or not likely to have been the predominant selection pressures in the natural environment. Collectively, these observations show that the reduced dataset is simply enriched for more positive SMEs (in this case, more neutral-positive SMEs) but retains similar levels of apparent idiosyncrasy. Furthermore, we speculate that adaptive mutations, when probed under adaptive conditions, have a higher tendency to exhibit magnitude epistasis as opposed to sign epistasis (in our dataset). Though we believe that deeper exploration of this pattern is outside of the scope of our study, we outline these arguments and hypothesis in LL. 372–381 to encourage further research.

2. Revisions requested by Reviewer #2:

- **Minor Revision #5:**

One suggestion was to connect this work to a substantial past record of work on intramolecular epistatic networks that contain the high-order interactions of amino acids and the connection of these networks to ligand-mediated signal propagation and allosteric communication. The authors have responded with a new paragraph describing statistical frameworks for analyzing coevolution of amino acids and the emergent concept of protein sectors. The authors should correct the discussion to be clear that the ideas of sectors came from direct statistical decompositions of empirical correlations (statistical coupling analysis (SCA)) and only later formalized in generative Potts models such as DCA. The ideas described by the authors are more or less correct, but the historical record should be made correct.

Response to Rev #5:

We thank the reviewer for highlighting this oversight. We have corrected the discussion to clarify that the concept of sectors originally emerged from experimental studies employing SCA and was later expanded upon with DCA. We also included an insightful review by O'Rourke *et al.* {PMID: 27441044} that outlines experimental and computational methods in the context of intramolecular network analysis. These changes can be found in LL. 413–418.

- **Minor Revision #6:**

A second (admittedly non-trivial) issue was to provide some rationale to explain an apparent inconsistency between the concept of pervasive epistasis along adaptive trajectories and the finding that background averaged epistasis in proteins can be sparse and yet near-complete with regard to predictive and even generative power. Indeed, how can we get away with a sparse representation that can generate proteins with fitness above a selection threshold if there is extensive epistasis in the paths that link such solutions? In this regard, it is not fully clear that the authors have addressed the central issue. In the opinion of this reviewer, it is not about the need to integrate structural and epistatic information. This is about the fact that adaptive trajectories collected with positive selection will focus adaptive mutations in highly epistatic units within proteins and that within such units, the effects of lower-order mutations will be highly context dependent (due to the existence of sparse but high-order epistatic terms not exposed in these works). Generative models learn approximate representations of these collective epistatic terms and can thus be sparse. The authors need not reconstruct these arguments in this current work, but at the least they can point out the apparent inconsistency as an issue for further work.

Response to Rev #6:

We apologize for the misunderstanding of the reviewer's previous comment. We originally aimed at providing a rationale for how predictive/generative models can be reconciled with biochemical/intramolecular network modelling in order to improve functional predictions, hence the commentary in LL. 433–445. Although in this discussion we do not address the central (and pivotal) issue described by the reviewer, we believe that our study provides novel research avenues that will be beneficial to researchers interested in utilizing structural data to improve existing predictive models. Thus, we have retained this discussion in the manuscript.

Indeed, the apparent discrepancy between the existence of pervasive epistasis (resulting in the poor predictive power of the biochemical model) and the relative success of predictive/generative models must be addressed. We fully agree with all the reviewer's remarks, and although we implicitly comment on the poor predictive power of the biochemical model in our discussion (LL. 382–386), we acknowledge that this contradiction must be addressed more explicitly. Thus, we have expanded this section addressing the reviewer's claim in LL. 386–397. This new explanation follows our discussion about the predictive power of the biochemical model, which it complements well, providing a stronger rationale for our observations.